# MPCACHE: MPC-Friendly KV Cache Eviction For Efficient Private LLM Inference

**Wenxuan Zeng**[1], **Ye Dong**[2], **Jinjin Zhou**[3], **Jin Tan**[3], **Lei Wang**[3]
**Tao Wei**[3], **Runsheng Wang**[1], **Meng Li**[1*]
[1]Peking University    [2]National University of Singapore    [3]Ant Group
zwx.andy@stu.pku.edu.cn, meng.li@pku.edu.cn

## Abstract

Private large language model (LLM) inference based on secure multi-party computation (MPC) achieves formal data privacy protection but suffers from significant latency overhead, especially for long input sequences. While key-value (KV) cache eviction and sparse attention algorithms have been proposed for efficient LLM inference in plaintext, they are not designed for MPC and cannot benefit private LLM inference directly. In this paper, we propose an accurate and MPC-friendly KV cache eviction framework, dubbed MPCACHE, building on the observation that historical tokens in a long sequence may have different effects on the downstream decoding. Hence, MPCACHE combines a look-once static eviction algorithm to discard unimportant KV cache and a query-aware dynamic selection algorithm to activate only a small subset of KV cache for attention computation. MPCACHE further incorporates a series of optimizations for efficient dynamic KV cache selection, including MPC-friendly similarity approximation, hierarchical KV cache clustering, and cross-layer index-sharing strategy. Extensive experiments demonstrate that MPCACHE consistently outperforms prior-art KV cache eviction baselines across different generation tasks and achieves $1.8 \sim 2.01\times$ and $3.39 \sim 8.37\times$ decoding latency and communication reduction on different sequence lengths, respectively. The code can be found here.

## 1 Introduction

Large language models (LLMs) have recently demonstrated remarkable ability in a wide range of applications such as document summarization [32, 57, 86], question answering [37, 15, 80], and dialogue systems [69, 11, 68]. However, LLM-based machine learning as a service (MLaaS) on the cloud has raised serious privacy concerns as the users are required to upload their prompts to the cloud, which may contain sensitive personal information. Meanwhile, the service provider is unwilling to offload the trained model to the user to protect the proprietary model weights. In recent years, secure multi-party computation (MPC)-based LLM inference has become a prevailing and mainstream research trend to address privacy concerns [53, 31, 19, 25, 60, 62, 94, 48], which enables the users and the cloud to conduct the LLM inference jointly, but nothing else can be derived beyond the final inference results.

However, MPC-based LLM inference illustrated in Figure 1(a) faces serious efficiency challenges, especially for long sequences. We profile the decoding efficiency of GPT-2 with the Secretflow framework [55] using recent 2-party computation (2PC) [53] and 3-party computation (3PC) protocols [19]. As can be observed in Figure 1(b) and (c), *attention computation dominates the latency and communication for both 2PC and 3PC protocols. Moreover, Softmax accounts for the majority of the*

---

*Corresponding author.

39th Conference on Neural Information Processing Systems (NeurIPS 2025).

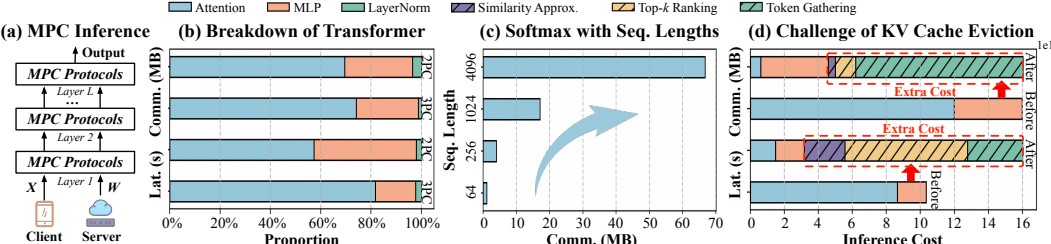

Figure 1: (a) MPC-based LLM inference. (b) Breakdown of decoding latency and communication with a sequence length of 512. Attention dominates the overhead for both 3PC and 2PC protocols. (c) The cost of Softmax scales with the sequence length rapidly. (d) Inference cost before and after KV cache eviction. Blocks in slash indicate the extra overhead introduced by eviction.

*overall cost due to its complex operators including max, exponential, and division, especially with an increasing sequence length.*

To reduce the cost of private LLM inference, previous works focus on developing more efficient MPC protocols [53, 19, 60, 31], replacing non-linear activation functions with more MPC-friendly operators [48, 40, 84], or directly modifying the model architecture [62]. However, they still incur significant overhead or require expensive finetuning or re-training, and cannot be directly applied to LLMs. Another line of works leverages key-value (KV) cache eviction (sparse attention) to reduce the number of tokens involved in the attention computation [91, 22, 50, 93, 89, 20]. Although they have demonstrated significant memory and computation reduction for plaintext LLM inference without the need of finetuning, they are not friendly to MPC. As shown in Figure 1(d), directly applying the dynamic KV cache eviction algorithm [47] incurs even more communication and latency overhead over the baseline model since it introduces expensive operators in MPC, including top-$k$ ranking, token gathering, etc, as elaborated in Section 3. Therefore, there is an urgent need for a training-free MPC-friendly algorithm designed for private LLM inference.

To overcome the heavy overhead of attention computation, we make the following observations that motivate our MPCACHE: 1) LLM attention maps are overall sparse for long input prompts, motivating us to perform static eviction and directly prune the KV cache of unimportant tokens; 2) LLM attention shows token-wise locality [49], motivating us to build an efficient clustering algorithm for dynamic selection of the KV cache; 3) LLM attention of adjacent layers shows similar patterns, motivating us to share the KV cache selection for adjacent layers to further improve efficiency. Our contributions can be summarized as follows:

- We observe the cost of MPC-based LLM inference mainly comes from attention computation and propose MPCACHE, an MPC-friendly KV cache eviction framework to reduce the LLM inference latency and communication.

- We identify the challenges when applying KV cache eviction to MPC. To tackle the problems, MPCACHE combines look-once static eviction and query-aware dynamic selection with a series of optimizations, including MPC-friendly similarity approximation, hierarchical KV cache clustering, and cross-layer index-sharing strategy.

- With extensive experiments, we demonstrate the performance of MPCACHE consistently exceeds the prior-art KV cache eviction algorithms across different generation tasks and achieves up to $2.01\times$ and $8.37\times$ decoding latency and communication, respectively.

## 2 Problem Formulation and Background

### 2.1 Problem Formulation

Generative LLM inference can be divided into prefill and decoding stages (refer to Appendix A.3). We formally describe the decoding process with KV cache eviction in Algorithm 1. The KV cache eviction policy, denoted as $\mathcal{P}$, aims to minimize the attention computation by only preserving a subset of tokens, which typically involves three steps: 1) $\mathcal{P}$ first computes the similarity between the query and key cache of previous tokens (line # 1); 2) $\mathcal{P}$ then ranks the previous tokens based

Table 1: Qualitative comparison with prior works of attention optimization.

| Representative Work | Category | Token Similarity Approximation | Top-$k$ Ranking | Token Gathering | Layer-wise Optimization | MPC Efficiency | Model Performance |
|---|---|---|---|---|---|---|---|
| [40, 84, 90] | Softmax Approximation | - | - | - | - | Training Required | Impractical for LLM |
| [75, 82, 79] | Learning-based | - | - | - | - | Training Required | High |
| [76, 3, 26] | Fixed-pattern | - | - | - | - | High | Low |
| [42, 71, 5] | Static | Accumulated Attn. | Usually Once | Token-wise | - | High | Low |
| [47, 67, 8] | Dynamic | Token-wise Similarity | Token-wise & Step-wise | Token-wise | - | Low | High |
| MPCACHE (ours) | Static+Dynamic | Hierarchical Clustering Cluster-wise Similarity | Cluster-wise & Step-wise | Cluster-wise | Cross-layer Index-sharing | High | High |

on the similarity score and applies the top-$k$ algorithm to determine the indices of relevant tokens (line # 2); 3) the KV cache is then retrieved based on the indices, denoted as token gathering (line # 3)[2], followed by sparse attention computation with the selected KV cache (line # 4). To compute the similarity in line # 1, existing works have used accumulated attention score of the historical tokens [50, 91, 93, 78, 89] or cosine similarity [47, 74].

KV cache eviction reduces the attention computation complexity from $\mathcal{O}(Td)$ to $\mathcal{O}(kd)$, where $T, d$ denote the sequence length and embedding dimension, respectively, and $k \ll T$. However, it introduces MPC-unfriendly operations, including similarity approximation, top-$k$ ranking, and token gathering, hindering its benefits in MPC-based LLM inference. Hence, the goal of this work can be summarized as

---

**Algorithm 1:** Formulation of KV cache eviction

**Input** : Query, key, and value cache $\mathbf{q}, \mathbf{K}, \mathbf{V}$.
**Output :** Sparse attention output $\mathbf{O}$

1 $\mathbf{sim} = \mathrm{Sim}(\mathbf{q}, \mathbf{K})$;                 ▷ Similarity approx.
2 $\mathbf{idx} = \mathrm{topk}(\mathbf{sim}, k = k)$;             ▷ Top-$k$ ranking
3 $\mathbf{K}', \mathbf{V}' = \mathbf{K}.\mathrm{gather}[\mathbf{idx}], \mathbf{V}.\mathrm{gather}[\mathbf{idx}]$;   ▷ Gathering
4 $\mathbf{O} = \mathrm{Softmax}(\mathbf{q} \cdot \mathbf{K}'^{\top}/\sqrt{d}) \cdot \mathbf{V}'$;   ▷ Sparse attention
5 **return** $\mathbf{O}$.

---

follows: ***"How can we design an MPC-friendly KV cache eviction algorithm $\mathcal{P}$ to minimize MPC-based LLM inference overhead without sacrificing LLM performance?"***

## 2.2 Background

**KV cache eviction.** There has been a surge in improving the efficiency of private LLM inference. Existing works focus on the protocol optimization [60, 19, 53] or replacing non-linear functions with MPC-friendly operators [40, 84, 18]. However, they either incur large overhead for long sequences or require expensive training. KV cache eviction has been widely explored for plaintext inference and can be mainly classified into 3 categories: 1) *fixed-pattern algorithms* like [76, 3] always keep the tokens in the same position, e.g., initial and recent tokens across decoding steps, lacking flexibility for different LLMs and contexts; 2) *static algorithms* like [91, 42, 22] discard tokens based on the accumulated attention scores of historical tokens, which are efficient but suffers from large performance degradation when the compression ratio is high; 3) *dynamic algorithms* like [74, 67, 29] compute the similarity between the current query and key cache for every decoding step, which is more accurate but requires repetitive token selection at each

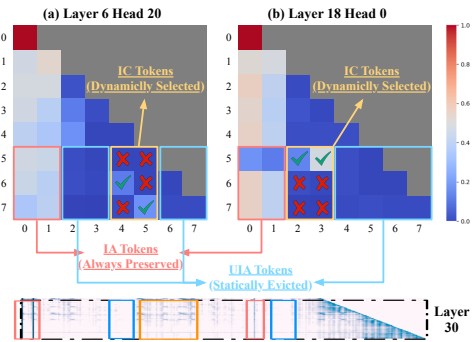

Figure 2: (Upper) token types in attention maps where ✓ means the token is selected and ✗ means the token is not selected. (Lower) three types can be observed in attention map with more tokens.

step. Recently, there are learning-based methods [75, 21] which rely on the training process. Different from prior works in Table 1, MPCACHE is a training-free and hybrid method and equipped with MPC-friendly optimizations, simultaneously achieving high efficiency and performance. We leave a detailed review of existing works in Appendix A.

---

[2]We describe the procedure and protocol of token gathering in Appendix B.4.

**MPC and threat model.** MPC [24] is a cryptographic technique recently developed to enable LLM inference while protecting the privacy of both data and model, as shown in Figure 1(a). In an MPC framework, to protect a certain tensor, it is often split into multiple secret shares and distributed across different parties involved in the computation [53, 19, 56, 31]. We adopt an *honest-but-curious* threat model and apply MPCACHE to both 2PC and 3PC protocols, which involve 2 parties and 3 parties in the computation, respectively. We refer interested readers to Appendix B, where the threat model and underlying protocols are more clearly explained.

## 3   Motivations and Challenges

In this section, we discuss the key observations that motivate the design of MPCACHE.

**Observation 1: the attention map of a long input sequence is usually sparse, and the KV cache of historical tokens demonstrates different impacts over the downstream decoding.**   We show the attention map of different heads and layers of LLaMA-2-7B in Figure 2 and leave visualizations of larger attention maps in Appendix C. From Figure 2, we can classify different tokens into 3 categories: 1) important to all tokens (IA in red box): the attention scores remain high for the entire column,

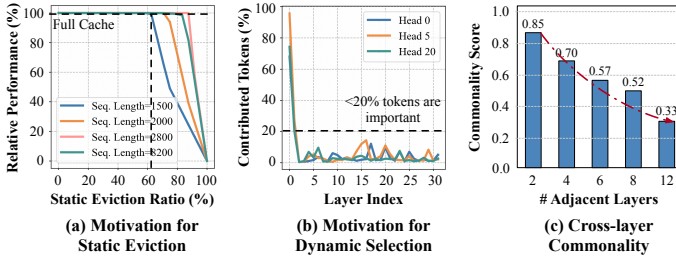

(a) Motivation for Static Eviction    (b) Motivation for Dynamic Selection    (c) Cross-layer Commonality

Figure 3: Motivating inspirations. (a) Statically evicting 60% tokens during the prefill stage still maintains the performance; (b) less than 20% tokens contribute to decoding; (c) cross-layer commonality among different numbers of adjacent layers.

e.g., 0th and 1st columns in Figure 2(a), indicating these tokens are important for the generation of all downstream tokens and hence, need to be always preserved; 2) un-important to all tokens (UIA in blue box): the attention scores remain low for the entire column, e.g., 2nd and 3rd columns in Figure 2(a), indicating these tokens can be discarded without impacting the downstream decoding; 3) important to certain tokens (IC in orange box): the attention scores vary for different tokens, e.g., 4th and 5th columns in Figure 2(a), indicating these tokens impact a subset of downstream tokens, and hence, cannot be directly pruned.

We verify the observations on LLaMA-2-7B with different input sequence lengths. As shown in Figure 3(a), almost 60% tokens can be statically evicted while preserving the LLM performance. While further pruning the remaining tokens starts to degrade the LLM performance, as shown in Figure 3(b), in each decoding step, only less than 20% of the remaining tokens contribute to the decoding. ***The above observation motivates us to statically evict the KV cache of UIA tokens and dynamically select a subset of IC tokens at each decoding step.***

**Observation 2: dynamic KV cache selection incurs non-negligible overhead in MPC.** While dynamic KV cache selection is accurate and reduces the attention computation cost, it incurs non-negligible overhead due to MPC-unfriendly operations. In Figure 1(d), we show the extra overhead when 5% tokens are dynamically selected. The MPC-unfriendly operations mainly include:

- Similarity computation (Algorithm 1 line # 1): cosine similarity is widely used, which requires computing the multiplication between the current query with the key cache of all previous tokens;

- Top-$k$ ranking (Algorithm 1 line # 2): to compute the indices of relevant tokens, top-$k$ is usually inevitable [91, 22, 93]. Unlike plaintext inference, top-$k$ ranking in MPC involves frequent comparison protocol, which incurs high latency and communication cost [63].

- Token gathering (Algorithm 1 line # 3): after the top-$k$ ranking, the KV cache of selected tokens is gathered based on the indices. Unlike plaintext inference, such gathering protocol in MPC is much more expensive since both KV cache and indices are ciphertexts. Therefore, as described in Algorithm 2, each index is first converted to a one-hot vector and then multiplied with the KV cache, requiring repetitive MPC-unfriendly equal protocols.

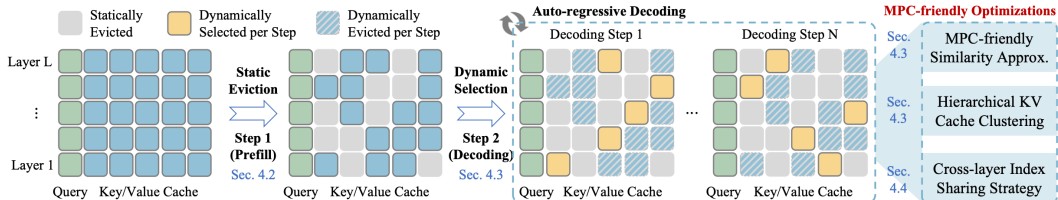

Figure 4: Overview of our proposed MPCACHE.

Inspired by token-wise locality [49, 95], *our key insight is to group the adjacent tokens into clusters*, which can reduce the complexity of dynamic selection in proportion to the cluster size. However, this introduces extra questions on how to measure the similarity between a cluster and the current query, how to build the cluster, etc, which is discussed in Section 4.3.

**Observation 3: adjacent layers share similar top-$k$ ranking of KV cache, providing an extra opportunity for efficiency optimization.** Due to the residual, we hypothesize adjacent layers may share a similar top-$k$ ranking of the KV cache. To verify the assumption, we define commonality score to measure the ratio of common top-$k$ indices of $m$ adjacent layers as

$$\frac{1}{k(L-m)} \sum_{l=1}^{L-m} \left| \bigcap_{i=l}^{l+m} \mathbf{idx}_i[:k] \right|,$$

where $\mathbf{idx}_i[:k]$ denotes the set of top-$k$ indices for $i$-th layer, $L$ is the number of layers, and $|\cdot|$ counts the number of elements in a set. As shown in Figure 3(c), adjacent layers demonstrate a high similarity of top-$k$ indices, which indicates the query tends to focus on the KV cache of the similar tokens. The score reduces when $m$ increases, which motivates us to share the indices of selected tokens among adjacent layers to trade off efficiency and performance.

# 4 MPCACHE: MPC-friendly LLM Inference

## 4.1 Overview of MPCACHE

**Overall design.** Driven by the observations, we propose an MPC-friendly KV cache eviction framework, dubbed MPCACHE as shown in Figure 4. It consists of two steps: 1) look-once static eviction during the prefill stage to discard the UIA tokens (Section 4.2); 2) query-aware dynamic selection during the decoding stage to select only a small subset of the remaining IC tokens for sparse attention (Section 4.3). A series of MPC-friendly optimizations are proposed to reduce the overhead of selection (Section 4.3 and 4.4). The pseudocode is shown in Algorithm 3 and the overall data flow is shown in Appendix H.

**Symbol definition.** For clarity, we summarize the symbols used in this paper. We define $L$ as the number of layers, $H$ as the number of attention heads, $T$ as the number of tokens, $d$ as the embedding dimension, $s$ as the cluster size, and $C$ as the number of clusters.

## 4.2 Step 1: Look-once Static KV Cache Eviction

To prune the KV cache of UIA tokens as observed in Section 3, we use static eviction once during the prefill stage. To measure the token importance and identify UIA tokens, we compute the attention map and then, accumulate the attention scores for each token. Similar to [42, 93], we find it is sufficient to only sum up the scores of the last 20% tokens in the prompt to compute the accumulated attention score. Then, we rank the accumulated attention scores to select the important tokens with the highest scores and discard the UIA tokens.

*Protocol complexity analysis.* Compared to the baseline computation of the prefill stage, static eviction only involves accumulating the attention scores, which are locally computed without any communication, and top-$k$ ranking. Because the static eviction is performed only once, the

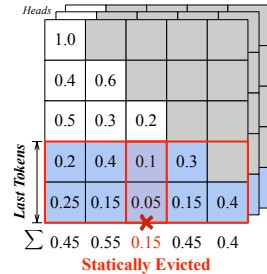

Figure 5: The illustration of static eviction.

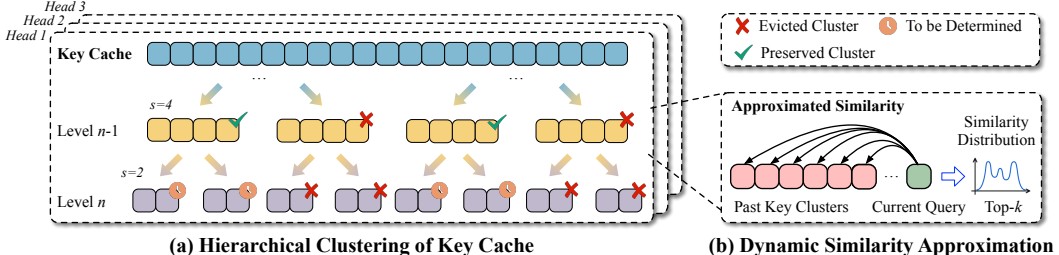

Figure 6: Hierarchical and dynamic KV cache clustering and selection procedure.

Table 2: The complexity analysis of token gathering protocol where $k_1 = 0.25T, k_2 = 0.25C$.

| | Bit Width | # Comparison | Lat. | Comm. | Example Lat. | Example Comm. |
|---|---|---|---|---|---|---|
| Baseline Protocol | $\log T$ | $T$ | $\mathcal{O}(T \log T)$ | $\mathcal{O}(k_1 T \log T)$ | 4.780s | 416.0MB |
| MPCACHE | $\log C$ | $C$ | $\mathcal{O}(C \log C)$ | $\mathcal{O}(k_2 C \log C)$ | 0.065s | 1.125MB |
| Improvement | $\frac{\log T}{\log C} \times$ | $\frac{T}{C} \times$ | $\frac{T \log T}{C \log C} \times$ | $\frac{k_1 T \log T}{k_2 C \log C} \times$ | 73.5× | 369.8× |

cost can be amortized by the entire generation process, and hence, becomes negligible. Meanwhile, with UIA tokens pruned, the efficiency of the dynamic selection can be improved for all decoding steps and overall efficiency.

### 4.3 Step 2: MPC-friendly Dynamic KV Cache Selection

To reduce the overhead of dynamic token selection as shown in Figure 1(c), we propose to group the KV cache of adjacent tokens into clusters in Figure 6(b). The most important question is *"how to aggregate the information of a cluster and measure the cluster importance accurately and efficiently?"*

**MPC-friendly similarity approximation with clustering.** A naive method for similarity approximation is to compute the average of the key cache within a cluster and directly compute the cosine similarity with the average. However, as shown in Figure 7, the naive approach incurs large performance degradation. *Our intuition is that the approximation should preserve the impact of important tokens as much as possible.* Hence, we use the maximum dot product between the query and the key cache cluster. Specifically, given a query $\mathbf{q} \in \mathbb{R}^{1 \times d}$, a key cache cluster of $s$ tokens $\mathbf{K}_c \in \mathbb{R}^{s \times d}$, the similarity can be designed as

$$\text{Sim}(\mathbf{q}, \mathbf{K}_c) = \max_{\mathbf{k} \in \mathbf{K}_c} \mathbf{q} \cdot \mathbf{k} = \max_{\mathbf{k} \in \mathbf{K}_c} \sum_{i=0}^{d-1} \mathbf{q}_i \mathbf{k}_i \leq \sum_{i=0}^{d-1} \max_{\mathbf{k} \in \mathbf{K}_c} \mathbf{q}_i \mathbf{k}_i. \tag{1}$$

We obtain the upper bound of similarity, and further have

$$\max_{\mathbf{k} \in \mathbf{K}_c} \mathbf{q}_i \mathbf{k}_i = \begin{cases} \mathbf{q}_i \max_{\mathbf{k} \in \mathbf{K}_c} \mathbf{k}_i & \text{if } \mathbf{q}_i \geq 0, \\ \mathbf{q}_i \min_{\mathbf{k} \in \mathbf{K}_c} \mathbf{k}_i & \text{if } \mathbf{q}_i < 0. \end{cases}$$

Define $\mathbf{r}^{\max}$ and $\mathbf{r}^{\min}$, where $\mathbf{r}_i^{\max} = \max_{\mathbf{k} \in \mathbf{K}_c} \mathbf{k}_i$ and $\mathbf{r}_i^{\min} = \min_{\mathbf{k} \in \mathbf{K}_c} \mathbf{k}_i$. Then, we have

$$\text{Sim}(\mathbf{q}, \mathbf{K}_c) \leq \sum_{i=0}^{d-1} \max(\mathbf{q}_i \mathbf{r}_i^{\max}, \mathbf{q}_i \mathbf{r}_i^{\min}). \tag{2}$$

*Protocol complexity analysis.* During the decoding stage, $\mathbf{r}_i^{\max}$ and $\mathbf{r}_i^{\min}$ of each cluster only need to be computed once. Hence, the computation cost can be amortized and become negligible. However, for each decoding step, we still need to compute $\mathcal{O}(LCd)$ multiplications, i.e., $\mathbf{q}_i \mathbf{r}_i^{\max}$ and $\mathbf{q}_i \mathbf{r}_i^{\min}$, as well as $\mathcal{O}(LCd)$ max operations in Equation (2), which still incur non-negligible overhead.

**Linearization and reordering.** To avoid the MPC-unfriendly max operation in Equation (2), we further propose to approximate the similarity as below:

$$\text{Sim}(\mathbf{q}, \mathbf{K}_c) \approx \sum_{i=0}^{d-1} \alpha \cdot \mathbf{q}_i \mathbf{r}_i^{\max} + (1 - \alpha) \cdot \mathbf{q}_i \mathbf{r}_i^{\min},$$

where $\alpha \in [0, 1]$ is a hyperparameter. As can be observed, when $\alpha = 1$, $\mathbf{q}_i \mathbf{r}_i^{\max}$ is always selected while $\mathbf{q}_i \mathbf{r}_i^{\min}$ is always selected when $\alpha = 0$. After the linearization, there is an opportunity to further reduce the multiplications by reordering the computation as

$$\sum_{i=0}^{d-1} \alpha \cdot \mathbf{q}_i \mathbf{r}_i^{\max} + (1 - \alpha) \cdot \mathbf{q}_i \mathbf{r}_i^{\min} = \sum_{i=0}^{d-1} \mathbf{q}_i \cdot (\alpha \mathbf{r}_i^{\max} + (1 - \alpha) \mathbf{r}_i^{\min}). \tag{3}$$

$\alpha \mathbf{r}_i^{\max}$ and $(1 - \alpha) \mathbf{r}_i^{\min}$ are first added up without introducing extra communication, and the multiplication with $\mathbf{q}_i$ is reduced by $2 \times$. Compared with the maximum dot product in Figure 7, our method significantly reduces the cost while maintaining the performance. We empirically choose $\alpha = 0.6$, and leave more discussions to Appendix F.3 and a theoretical analysis to Appendix G.

*Protocol complexity analysis.* MPCACHE reduces the number of max operations from $\mathcal{O}(LCd)$ to 0 and reduce the multiplication complexity by $2 \times$. It is worth noting that clustering also benefits the token gathering protocol: 1) the number of equal protocol in one-hot vector conversion is reduced by $\frac{T}{C} \times$; 2) the bit width of one-hot vector is reduced by $\frac{\log T}{\log C} \times$. Table 2 shows an example of selecting top-$25\%$ tokens with $T = 1024, C = 64$, and the overhead is drastically reduced.

**Hierarchical KV cache clustering.** Another question is *"how to build the KV cache cluster efficiently?"* Since larger cluster sizes have higher efficiency at the cost of worse performance, we propose to trade off the selection overhead and performance. Inspired by hierarchical reinforcement learning [77], we propose to cluster the KV cache of adjacent tokens with a hierarchical structure as shown in Figure 6(a) that performs coarse-grained (larger cluster size) to fine-grained (smaller cluster size) selection progressively. Generally, we divide the KV cache into $n$

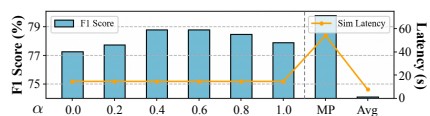

Figure 7: Comparison among maximum dot product (MP), average, and our method with different $\alpha$'s on TriviaQA.

levels and progressively select the clusters level by level. At the fine-grained level, we only need to select from the remaining clusters, thereby reducing the selection complexity. Hierarchical structure, including the cluster size and selection ratios at each level, can influence the performance-efficiency trade-off. We discuss the trade-off in Section 5.4 and setups in Appendix F.1.

### 4.4 Cross-layer Index-sharing Strategy

Based on the observation that adjacent layers share similar top-$k$ ranking of KV cache, we propose a cross-layer index-sharing strategy that enables adjacent layers to share the same selected token indices to further reduce the cost of dynamic selection. Since two adjacent layers show the highest commonality score in Figure 3(c), we choose to share the indices between two adjacent layers. In Figure 8, we observe the first two layers have a low commonality score while other layers have higher scores due to the residual, so we do not apply sharing to the first two layers. Cross-layer index-sharing effectively reduces the

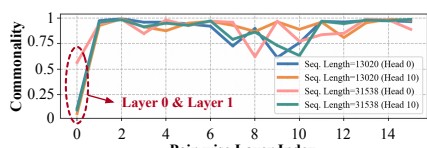

Figure 8: Commonality score between two adjacent layers.

extra overhead introduced by dynamic selection. We discuss how the number of adjacent layers affects the trade-off in Section 5.4.

### 4.5 Security Analysis of MPCACHE

MPCACHE is built upon established cryptographic building blocks from 2PC [53] and 3PC [19] that were already proven secure. Our introduced top-$k$ protocol follows [31]. Since our other optimizations do not introduce new protocols, but combine them to support efficient KV cache eviction, MPCACHE follows naturally from the security of these building blocks. MPCACHE assumes all parties are aware of the model architecture and number of pruned tokens, which is consistent with [88, 41, 38]. We argue that it does not compromise the client's data, nor does it enable the client to access the model's parameters. Also, the shared indices between layers are ciphertexts, which won't reveal any private information.

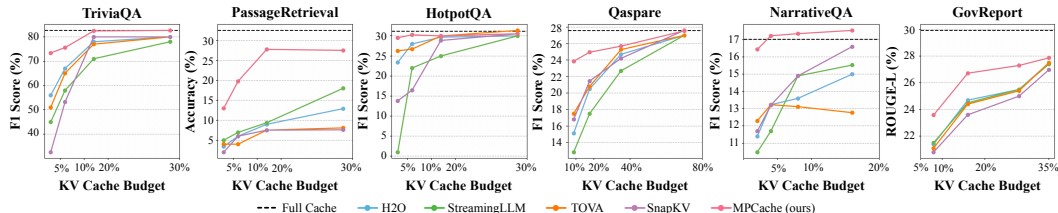

Figure 9: Comparison with prior-art fixed-pattern and static eviction algorithms on LongBench.

Table 3: Comparison with prior-art dynamic KV cache eviction algorithms with different budgets.

| Dataset | Cache Budget | InfLLM | | LongCache | | MPCACHE (ours) | |
|---------|--------------|--------|--------|-----------|--------|----------------|--------|
| | | Performance (%)↑ | Latency (s)↓ | Performance (%)↑ | Latency (s)↓ | Performance (%)↑ | Latency (s)↓ |
| HotpotQA | Full | 31.16 | 75.52 | 31.16 | 75.52 | 31.16 | 75.52 |
| | 5% | 28.20 | 51.64 (1.30×) | 24.31 | 89.46 (2.24×) | 30.27 | 39.85 |
| | 10% | 29.01 | 68.04 (1.28×) | 24.69 | 123.1 (2.30×) | 30.05 | 53.32 |
| TriviaQA | Full | 82.67 | 75.52 | 82.67 | 75.52 | 82.67 | 75.52 |
| | 5% | 75.65 | 51.64 (1.38×) | 59.85 | 89.46 (2.39×) | 75.61 | 37.37 |
| | 10% | 82.75 | 68.04 (1.34×) | 60.56 | 123.1 (2.43×) | 82.45 | 50.75 |
| NarrQA | Full | 17.02 | 75.52 | 17.02 | 75.52 | 17.02 | 75.52 |
| | 5% | 12.80 | 47.74 (1.32×) | 14.65 | 86.42 (2.39×) | 17.23 | 36.13 |
| | 10% | 13.74 | 63.49 (1.28×) | 15.69 | 121.4 (2.45×) | 17.35 | 49.46 |
| PassageR | Full | 32.50 | 75.52 | 32.50 | 75.52 | 32.50 | 75.52 |
| | 5% | 6.161 | 51.64 (1.15×) | 21.42 | 89.46 (1.99×) | 19.75 | 44.82 |
| | 10% | 8.872 | 68.04 (1.16×) | 24.92 | 123.1 (2.10×) | 27.75 | 58.47 |
| Qasper | Full | 27.58 | 75.52 | 27.58 | 75.52 | 27.58 | 75.52 |
| | 8% | 20.53 | 64.52 (1.45×) | 24.53 | 136.9 (3.08×) | 23.86 | 44.39 |
| | 16% | 23.90 | 72.84 (1.33×) | 26.07 | 225.9 (4.12×) | 24.95 | 54.77 |

* (a×) means MPCACHE achieves a× efficiency improvement over the baselines.

## 5 Empirical Evaluation

### 5.1 Experimental Setups

**Models, datasets, and baselines.** Our experiments are conducted on LLaMA-2, LongChat-7B-V1.5-32K, and LLaMA-3.1-8B-Instruct on LongBench [2], XSUM [57], and Needle-in-a-Haystack [23]. Refer to Appendix F.1 for details. We choose prior-art static and dynamic KV cache eviction baselines, including H2O [91], StreamingLLM [76], TOVA [58], SnapKV [42], InfLLM [74], and LongCache [47]. We leave the details of baselines to Appendix F.2.

**Experimental setups.** For performance evaluation, our experiments are conducted based on LongBench on an NVIDIA A100 80GB GPU. For efficiency evaluation, we use Secretflow (SPU V0.9.1) [55] and follow the 3PC protocols of PUMA [19]. The latency is evaluated under the LAN setup. We evaluate the efficiency using the architecture of GPT-2 and LLaMA-2. We emphasize that the eviction ratio is determined based on the expected sequence length and the latency that can be tolerated

Table 4: Evaluation on XSUM task.

| Budget | 10% | | 5% | |
|--------|-----|-----|-----|-----|
| Model | 7B↑ | 13B↑ | 7B↑ | 13B↑ |
| Full Cache | 11.90 | 13.60 | 11.90 | 13.60 |
| H2O | 10.50 | 13.24 | 4.886 | 9.081 |
| MPCACHE (ours) | 11.10 | 13.44 | 10.08 | 13.08 |

by the server and the client in a realistic scenario. We explain the details of baselines, setups, and how to securely determine the hyper-parameters in Appendix F.1.

### 5.2 Performance Evaluation

In Figure 9 and Table 3, we comprehensively compare MPCACHE with prior-art KV cache eviction methods on LongChat-7B-V1.5-32K, and we make the following observations: **1) comparison with fixed-pattern and static methods.** MPCACHE consistently outperforms prior-art baselines across different datasets. MPCACHE shows decent scalability to different KV cache budgets. For example, on HotPotQA and NarrativeQA, MPCACHE achieves comparable performance as full cache, even only ∼5% KV cache preserved; **2) comparison with dynamic methods.** MPCACHE achieves comparable and even better performance compared with InfLLM and LongCache. On NarrativeQA,

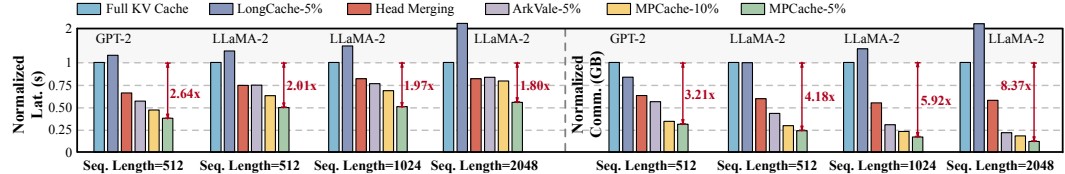

Figure 10: Evaluation on per-token generation latency and communication compared with LongCache [47], head merging [62], and ArkVale [8]. Note that 5% and 10% mean the final KV cache budget.

MPCACHE achieves $1.32\times$ and $2.39\times$ latency reduction with a higher F1 score compared with InfLLM and LongCache, respectively; **3) scalability.** We combine static and dynamic methods on LLaMA-2-13B in Table 4 and LLaMA-3.1-8B-Instruct in Table 5, demonstrating the superior performance of our method.

Table 5: Extension to LLaMA-3.1-8B-Instruct on LongBench with an average KV cache budget of 2048 and 1024. The best result is highlighted in **bold** and the second best in underline.

| Budget | Method | NarrativeQA | Qasper | HotpotQA | 2WikiMQA | MuSiQue | TREC | SAMSum | TriviaQA | QMSum | PR-en | MF-en |
|---|---|---|---|---|---|---|---|---|---|---|---|---|
| | Full Cache | 30.21 | 45.52 | 55.53 | 46.71 | 31.34 | 72.50 | 43.86 | 91.74 | 25.20 | 99.50 | 54.94 |
| 2048 | StreamingLLM | 27.40 | 30.77 | 49.23 | 44.66 | 24.31 | 67.50 | 42.49 | 90.98 | 21.67 | 87.00 | 37.85 |
| | DuoAttention | 25.61 | 42.31 | 52.36 | 42.14 | 28.17 | 66.00 | 43.36 | 89.93 | 22.11 | 98.50 | **55.53** |
| | SnapKV | 28.53 | 39.13 | 54.32 | 46.59 | 28.48 | 55.55 | 43.10 | **92.04** | 23.92 | 99.50 | 53.91 |
| | Quest | 28.00 | 45.55 | 53.73 | 44.76 | 29.82 | 68.56 | **44.05** | 90.90 | 24.91 | 99.50 | 55.40 |
| | MPCACHE | **30.17** | **46.11** | **55.21** | **46.61** | **30.49** | 69.50 | 43.67 | 91.53 | 24.83 | 99.50 | 53.41 |
| | Full Cache | 30.21 | 45.52 | 55.53 | 46.71 | 31.34 | 72.50 | 43.86 | 91.74 | 25.20 | 99.50 | 54.94 |
| 1024 | StreamingLLM | 26.64 | 27.48 | 47.31 | 42.03 | 24.17 | 63.50 | 42.76 | 88.84 | 21.31 | 88.00 | 35.59 |
| | DuoAttention | 24.12 | 30.41 | 45.34 | 39.45 | 24.49 | 45.53 | 42.54 | 76.36 | 21.24 | 98.50 | 41.11 |
| | SnapKV | 26.51 | 33.14 | 54.42 | 43.34 | 27.74 | 43.01 | 40.52 | 90.02 | 23.10 | 99.50 | 49.10 |
| | Quest | 15.45 | 44.42 | 48.90 | 40.81 | 22.93 | 57.03 | 37.90 | 70.00 | 21.04 | 99.00 | 49.84 |
| | MPCACHE | **29.47** | **46.20** | **55.38** | **46.55** | **31.14** | 69.50 | 43.02 | 91.84 | 24.77 | 99.50 | **52.70** |

## 5.3 Efficiency Evaluation

In Figure 10, we benchmark the decoding efficiency with different sequence lengths ranging from 512 to 2048 and a static eviction ratio of 70%. We compare MPCACHE with full KV cache, LongCache, head merging [62], and ArkVale [8] with 3PC protocols. From the results, we make the following observations: 1) compared with full KV cache, MPCACHE achieves $1.59 \sim 2.01\times$, $1.46 \sim 1.97\times$, and $1.26 \sim 1.8\times$ latency reduction and $3.39 \sim 4.18\times$, $4.33 \sim 5.92\times$, and $5.51 \sim 8.37\times$ communication reduction with different sequence lengths, respectively; 2) compared with LongCache which performs dynamic token-wise selection, MPCACHE even achieves $3.85\times$ and $19.47\times$ latency and communication reduction, respectively; 3) MPCACHE shows higher efficiency than ArkVale due to our optimizations. We further discuss 2PC protocol [53] in Appendix F.3.

## 5.4 Ablation Studies of MPCACHE

**Effectiveness of different optimizations.** In Figure 11, we demonstrate the effectiveness of our proposed optimizations by adding them step by step on LLaMA-2-7B with a sequence length of 1024 and static eviction ratio of 70%. We make the following observations: 1) directly applying

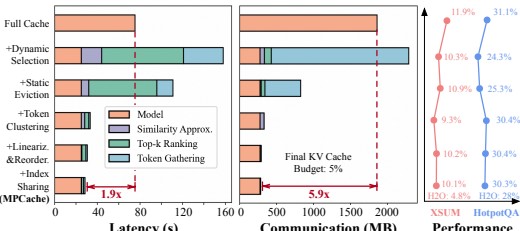

Figure 11: Step-by-step ablation study.

Table 6: Comparisons among different hierarchical structures.

| Level 1 Coarse-grained | Level 2 Fine-grained | F1 Score (%) | Comm. (MB) |
|---|---|---|---|
| $s32(0.9)$ | $s16(0.22)$ | 29.6 | 163.5 |
| $s32(0.7)$ | $s16(0.28)$ | 30.1 | 144.0 |
| $s32(0.5)$ | $s16(0.40)$ | 30.2 | 140.2 |
| $s32(0.3)$ | $s16(0.67)$ | 29.2 | 108.8 |
| $s64(0.9)$ | $s16(0.22)$ | 29.5 | 158.1 |
| $s64(0.7)$ | $s16(0.28)$ | 29.3 | 110.1 |
| $s64(0.5)$ | $s16(0.40)$ | 29.1 | 104.9 |
| $s64(0.3)$ | $s16(0.67)$ | 29.0 | 69.12 |

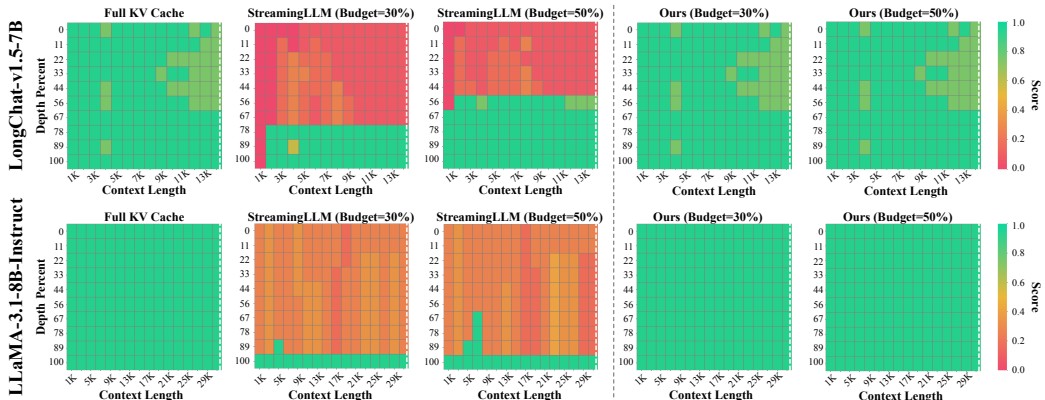

Figure 13: Needle-in-a-Haystack. The x-axis denotes the length of the context ("haystack") and the y-axis indicates the position where the "needle" (a short prompt) is inserted within the context.

dynamic selection, e.g., LongCache to private LLM inference does not provide the expected efficiency improvement and even increases both latency and communication; 2) after static eviction, latency and communication of dynamic selection are reduced by $1.42\times$ and $2.76\times$, respectively; 3) our MPC-friendly optimizations, including clustering, linearization, reordering, and cross-layer index-sharing further reduce the overhead introduced by dynamic selection without sacrificing the model performance; 4) MPCACHE eventually achieves $1.9\times$ and $5.9\times$ latency and communication reduction, respectively, and achieves high performance.

**Effect of hierarchical structures.** To trade off the model performance and dynamic selection overhead, we evaluate different hierarchical structures with a dynamic selection ratio of 20%. We choose different cluster sizes $s$ and selection ratios at different levels. From Table 6, we find that when $s$ increases or the coarse-grained selection ratio decreases, the overhead decreases while the performance exhibits a downward trend. Moreover, appropriate coarse-grained selection may help improve the overall performance, e.g., the ratio changes from 90% to 50% with $s = 32$;

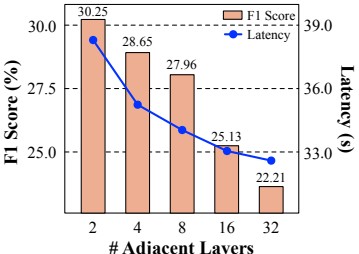

Figure 12: Effect of the number of adjacent layers.

**Effect of the number of adjacent layers for cross-layer index-sharing.** In response to Section 3, we evaluate the efficiency-performance trade-off of the number of adjacent layers on HotpotQA in Figure 12. As observed, when the number of adjacent layers increases, the latency is reduced while the performance degradation.

**Benchmark on Needle-in-a-Haystack.** We benchmark our similarity approximation algorithm on Needle-in-a-Haystack as shown in Figure 13. MPCACHE demonstrates comparable performance with full KV cache and outstanding performance compared with StreamingLLM.

**Additional results.** We present the influence of $\alpha$, discussion on 2PC protocol, comparison with average-based similarity in Appendix F.

## 6 Limitation and Conclusion

MPCACHE focuses on MPC inference, which still incurs non-negligible overhead compared to the plaintext counterpart. For certain tasks, such as those requiring the model to recall information from tokens considered unimportant, static eviction may fall short, which is a direction for future research.

In this work, we propose an MPC-friendly KV cache eviction framework named MPCACHE, that enables accurate and efficient private LLM inference. MPCACHE systematically combines static eviction and dynamic selection. To reduce the overhead of dynamic selection, we propose a series of MPC-friendly optimizations, including efficient similarity approximation, hierarchical KV cache clustering, and cross-layer index-sharing. Extensive evaluations demonstrate that MPCACHE consistently outperforms prior-art KV cache eviction baselines across different generation tasks and significantly reduces both latency and communication.

## Acknowledgements

This work was supported in part by NSFC under Grant 62495102, Grant 92464104, and Grant 62341407, in part by Beijing Municipal Science and Technology Program under Grant Z241100004224015, and in part by Ant Group through CCF-Ant Research Fund.

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

# A Detailed Background and Related Works

## A.1 Private LLM Inference

Recently, private LLM inference has attracted an increasing amount of research attention. Iron [27] uses coefficient encoding to compute homomorphic encryption (HE)-based convolution layers efficiently and uses MPC to compute non-linear layers. PUMA [19] proposes a series of 3PC protocols for both linear and non-linear functions to support private LLM inference, even under the scale of LLaMA-7B. BumbleBee [53] proposes HE-based protocols that enable the multiplication of large matrices and efficient protocols for non-linear functions similar to PUMA. CipherGPT [31] uses subfield vector oblivious linear evaluation (sVOLE) to reduce the communication of MatMuls significantly. BOLT [60] proposes a baby-step giant-step (BSGS) strategy that reduces the number of rotations on ciphertexts. SIGMA [25] achieves private GPT inference with function secret sharing (FSS) and accelerates the computation on GPUs. PermLLM [94] proposes an efficient protocol for non-linear functions based on the random permutation. However, they mainly focus on protocol optimization, and still incur significant overhead, especially on long sequences.

There are also works directly replacing expensive non-linear functions in Transformers, e.g., Softmax and GeLU with MPC-friendly operations. For instance, MPCFormer [40] and Secformer [54] simplify Softmax by replacing exponential with an MPC-friendly quadratic function. MPCViT [84] replaces exponential in Softmax with ReLU and selectively uses scaling attention using neural architecture search (NAS). SAL-ViT [90] introduces external attention while PriViT [18] uses squaring function to replace Softmax. RNA-ViT [7] and Power-Softmax [96] use high-order polynomial approximation. [48] directly uniformly replaces GeLU with ReLU and replaces exponential in Softmax with ReLU. THE-X [9] approximates Softmax using an estimation model. Although these methods achieve high inference efficiency, they cannot avoid finetuning or re-training to preserve the model performance, making LLM development impractical. In conclusion, the above works still suffer from heavy overhead. Moreover, these works always handle full-length contexts during the LLM generation, incurring large latency and communication when handling long sequences. To solve this problem, our work aims to compress the KV cache in attention without training. Our proposed method can also be applied to different protocol frameworks for efficiency improvement.

## A.2 KV Cache Compression

When tackling the LLM generation tasks, especially in long-context scenarios, the KV cache in the attention module becomes the most significant bottleneck due to the increasing sequence length. Therefore, how to effectively reduce the size of the KV cache is a high priority. System-level optimizations such as FlashAttention [14], FlashAttention-2 [13], FlashAttention-3 [65], and PagedAttention [39] have been proposed to alleviate the problem. Meanwhile, many recent research efforts have been devoted to algorithm-level optimizations. For example, ***quantization*** methods [30, 87, 35, 28, 51] have been proposed to compress KV cache to $1 \sim 4$ bits, ***low-rank decomposition*** methods [64, 6, 92, 85] project the KV cache into low-rank space. In this work, we follow FlashAttention [14] to save the GPU memory and focus on another research line of algorithm-level optimization called ***KV cache eviction***, which is designed to reduce the number of tokens and enable sparse attention without extra training.

KV cache eviction can be mainly categorized into 3 classes: 1) *fixed-pattern algorithm:* the position of important tokens is pre-defined before inference and remains consistent across decoding steps. However, this algorithm is not flexible for different LLMs and contexts [76, 3, 83]; 2) *static algorithm:* tokens are statically discarded and cannot be recovered in the subsequent decoding steps. This algorithm is usually efficient but suffers from significant performance degradation when the compression ratio is high [91, 50, 22, 42, 89, 78, 93, 10, 71, 33, 58]; 3) *dynamic algorithm:* tokens are dynamically selected across different decoding steps. This algorithm is much more flexible but the dynamic selection usually involves more expensive operations [74, 47, 67, 8]. We quantitatively compare existing methods in Table 1.

Here, we introduce recent works of KV cache eviction. StreamingLLM [76] proposes to keep a few initial tokens along with the recent tokens to recover the long-context performance. RazorAttention [66] theoretically analyzes the scope of effective attention vision for each head. Scissorhands [50], H2O [91], ALISA [93], spAtten [72], SnapKV [42] and LOOK-M [71], and TOVA [58] use the accumulated attention score of the historical tokens to select a small subset of KV cache. FastGen

[22] and MInference 1.0 [33] propose to allocate different eviction policies for different heads based on the sparsity pattern of the prompt. PyramidKV [89], PyramidInfer [78], and SqueezeAttention [73] consider allocating different KV cache budgets for different layers. InfLLM [74] and LongCache [47] propose to dynamically select tokens based on the relationship between the current query and the key cache of previous tokens. RetrievalAttention [44] establishes connections from the query to its nearest keys and the decoding query can first search its nearest query and then obtain the most relevant key vectors. LazyLLM [20] introduces an aux cache to enable selective KV cache eviction. Keyformer [1] finds that the distribution after token pruning becomes uneven and proposes to smooth the distribution. Squeezed Attention [29] and ClusterKV [45] proposes to cluster KV cache based on the semantic information. HashAttention [16] and HashEvict [46] recently design hash functions to enable efficient KV cache eviction. To get rid of the dependency of similarity computation, SirLLM [81] uses token entropy while [17] uses the L2-norm of the key cache to measure the token importance.

However, these works are not designed or optimized for MPC since they either statically discard tokens that cause significant performance degradation or dynamically select tokens that introduce more complex and MPC-unfriendly operations. Recently, learning-based methods [75, 21] have also emerged, which require training to automatically explore the sparsity patterns and thus, are not the focus of this work.

### A.3 Generative LLM Inference in Autoregressive-style

The generative LLM inference procedure is generally in an autoregressive style such as GPT-2 [61] and LLaMA [70], and mainly consists of two stages: 1) the prefill (prompt) stage and 2) the decoding (generation) stage.

**Prefill stage.** The prefill stage serves as the first step of generation. LLM takes a prompt sequence as input and generates key-value (KV) caches for each layer. The attention can be computed as

$$\mathbf{O}_{\text{prompt}} = \text{Softmax}(\mathbf{Q}_{\text{prompt}} \cdot \mathbf{K}_{\text{prompt}}^\top / \sqrt{d}) \cdot \mathbf{V}_{\text{prompt}}, \tag{4}$$

where $\mathbf{Q}_{\text{prompt}} \in \mathbb{R}^{H \times T \times d}$ denotes the query and $\mathbf{K}_{\text{prompt}} \in \mathbb{R}^{H \times T \times d}$, $\mathbf{V}_{\text{prompt}} \in \mathbb{R}^{H \times T \times d}$ denote the key and value cache, respectively. After the prefill stage, the KV cache is generated as $\mathbf{K}_{\text{cache}} \leftarrow \mathbf{K}_{\text{prompt}}$ and $\mathbf{V}_{\text{cache}} \leftarrow \mathbf{V}_{\text{prompt}}$. KV cache retains previously computed key-value pairs, eliminating the need for costly re-computation of previous key and value vectors [59]. Note that each layer is equipped with its unique KV cache and the generated KV cache is the foundation for the dowmstreaming decoding stage.

**Decoding stage.** The decoding stage uses and updates the stored KV cache to generate new tokens step-by-step. First, the KV cache is updated by concatenating new $\mathbf{k} \in \mathbb{R}^{H \times 1 \times d}$ and $\mathbf{v} \in \mathbb{R}^{H \times 1 \times d}$ as

$$\mathbf{K}_{\text{cache}} \leftarrow [\mathbf{K}_{\text{cache}} || \mathbf{k}], \ \mathbf{V}_{\text{cache}} \leftarrow [\mathbf{V}_{\text{cache}} || \mathbf{v}], \tag{5}$$

where $[\cdot || \cdot]$ denotes tensor concatenation. Therefore, the attention can be computed as

$$\mathbf{o}_{\text{dec}} = \text{Softmax}(\mathbf{q}_{\text{dec}} \cdot \mathbf{K}_{\text{cache}}^\top / \sqrt{d}) \cdot \mathbf{V}_{\text{cache}}, \tag{6}$$

where $\mathbf{q}_{\text{dec}} \in \mathbb{R}^{1 \times d}$ denotes the current query. The attention output $\mathbf{o}_{\text{dec}} \in \mathbb{R}^{1 \times d}$ is then sent to the multi-layer perceptron (MLP) layer for the subsequent computation.

**Explanation about the use of KV cache.** Although previous works overlook the use of KV cache [19, 53, 52], KV cache is already a fundamental component for existing generative LLMs to guarantee the inference correctness. If we do not use KV cache, the model has to re-compute all the previous tokens at each decoding step, significantly increasing the overhead. Hence, we apply KV cache to private LLM inference by default, and our aim is to reduce the computation associated with KV cache size for better efficiency. We also show the generation efficiency without KV cache in Section 5 and Appendix F.

## B  MPC Protocol Descriptions

### B.1  Threat Model and Security

Consistent with previous works [56, 40, 19], MPCACHE adopts an *honest-but-curious* (a.k.a., semi-honest) security model in honest-majority [43] where parties follow the protocol specifications but

may also try to learn more from the information than allowed. In our threat model, we assume all the parties are aware of the LLM architecture and number of pruned tokens, which is consistent with HEPrune [88], Seesaw [41], SENet [38], SNL [12], etc. We argue that this information does not compromise the client's data or inference results, nor does it enable the client to access the model's parameters.

## B.2 2PC Protocol

We follow the 2PC protocols proposed in BumbleBee [53]. The protocols are built based on the 2-out-of-2 additive secret sharing (SS), where secret value $x \in 2^\ell$ is shared by two random values $x_0, x_1 \in 2^\ell$ such that $x = x_0 + x_1 \pmod{2^\ell}$, and party $P_i$ gets $x_i$ (denoted as $[\![x]\!]$). SS supports both addition and multiplication on the secret shares. Without special declaration, we compute in $2^\ell$ and omit $\pmod{2^\ell}$ for brevity. In the case of $\ell > 1$ (e.g., $\ell = 64$) which support arithmetic operations (e.g., $+$, $-$, and $\cdot$), we refer to this type as *arithmetic secret sharing*. *Boolean secret sharing* refers to $\ell = 1$ where $(+, -)$ and $\cdot$ are replaced by bit-wise $\oplus$ and $\wedge$, respectively.

- *Addition.* $[\![x + y]\!]$ can be computed as $(x_0 + y_0, x_1 + y_1)$, where $P_i$ can compute its share locally.
- *Multiplication.* We write the multiplication of two shared values as $[\![xy]\!] = (x_0 + x_1)(y_0 + y_1) = x_0y_0 + x_1y_1 + x_0y_1 + x_1y_0$ where two cross terms $x_0y_1, x_1y_0$ can be computed using HE.

[53] uses HE scheme that is based on ring learning-with-error (RLWE). For more details about the 2PC protocol, please refer to [53, 55].

## B.3 3PC Protocol

We follow the 3PC protocols proposed in PUMA [19]. The protocols are built based on the 2-out-of-3 replicated secret sharing (RSS), where a secret value $x \in 2^\ell$ is shared by three random values $x_0, x_1, x_2 \in 2^\ell$ such that $x = x_0 + x_1 + x_2 \pmod{2^\ell}$, and party $P_i$ gets $(x_i, x_{i+1})$ (denoted as $[\![x]\!]$).

Let $(c_1, c_2, c_3)$ be public constants, and $([\![x]\!], [\![y]\!])$ be two secret-shared values. The secure addition and multiplication procedures are as follows:

- *Addition.* $[\![c_1x + c_2y + c_3]\!]$ can be computed as $(c_1x_0 + c_2y_0 + c_3, c_1x_1 + c_2y_1, c_1x_2 + c_2y_2)$, where $P_i$ can compute its share locally. When $(c_1 = 1, c_2 = 1, c_3 = 0)$, we get $[\![x + y]\!]$.
- *Multiplication.* Parties follow steps: i) first, $P_i$ computes $z_i = x_iy_i + x_{i+1}y_i + x_iy_{i+1}$ locally; ii) parties then perform *re-sharing* by letting $P_i$ sends $z_i' = \alpha_i + z_i$ to $P_{i-1}$, where $\alpha_0 + \alpha_1 + \alpha_2 = 0$ ($P_i$ can generate $\alpha_i$ using pseudorandom generators with negligible overhead as [56]); iii) finally, $\{(z_0', z_1'), (z_1', z_2'), (z_2', z_0')\}$ form the 2-out-of-3 replicated secret shares of $[\![xy]\!]$.

For more details about the 3PC protocol, please refer to [56, 55].

## B.4 Token Gathering Protocol

Token gathering is used to retrieve tokens in the KV cache based on the indices, which has the same functionality as $\mathrm{torch.gather(tensor, indices)}$ in PyTorch programming. We illustrate the overall procedure in Figure 14.

For brevity, in Algorithm 2, we show the pipeline that retrieves one token from the key cache (we also omit the head dimension for simplification). The first step is converting a ciphertext index $[\![\mathbf{id}]\!]$ into a ciphertext one-hot vector $[\![\mathbf{o}]\!] \in \mathbb{R}^{1 \times T}$ based on equal protocol $\Pi_{\mathrm{Equal}}$, where $T$ denotes the number of tokens. Given a ciphertext key cache $[\![\mathbf{K}]\!] \in \mathbb{R}^{T \times d}$, where $d$ denotes the embedding dimension, multiplying $[\![\mathbf{o}]\!]$ with $[\![\mathbf{K}]\!]$ (matrix-vector multiplication $\Pi_{\mathrm{MatVec}}$) can generate an output with a dimension of $1 \times d$, which is the retrieved token. To extend the case to retrieve $m$ tokens, we concatenate $m$ one-hot vectors to form a matrix $[\![\mathbf{O}]\!] \in \mathbb{R}^{m \times T}$, and then multiply $[\![\mathbf{O}]\!]$ with $[\![\mathbf{K}]\!]$ (matrix-matrix multiplication $\Pi_{\mathrm{MatMul}}$) to generate an output with a dimension of $m \times d$, which is the retrieved $m$ tokens.

Note that token gathering protocol is also used on the value cache, and its indices are consistent with that of the key cache.

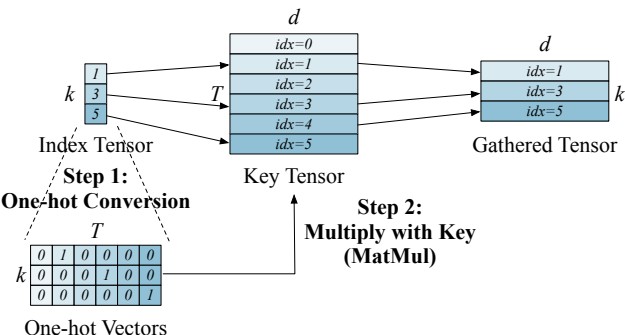

Figure 14: Illustration of token gathering procedure where we give an example of retrieving 3 tokens from 6 tokens.

---

**Algorithm 2:** Token gathering protocol $\Pi_{\text{Gather}}$ for retrieving one token

**Input** : A ciphertext key cache $[\![\mathbf{K}]\!] \in \mathbb{R}^{T \times d}$ and a ciphertext index $[\![\mathbf{id}]\!]$.
**Output**: Key cache $[\![\mathbf{K}]\!]' \in \mathbb{R}^{1 \times d}$ with the selected token.

1 **for** $i \in [0, \ldots, T-1]$ **do**
2  $\quad$ Parties jointly generate the one-hot vector as $[\![\mathbf{o}[i]]\!] = \Pi_{\text{Equal}}([\![\mathbf{id}]\!], i)$;
3 Parties jointly compute the retrieved key cache as $[\![\mathbf{K}]\!]' = \Pi_{\text{MatVec}}([\![\mathbf{o}]\!], [\![\mathbf{K}]\!])$;
4 **return** $[\![\mathbf{K}]\!]'$.

---

## C  Observation from Pattern Discovery of Large Attention Maps

It is sufficient to use a few tokens within the observation window to distinguish the attention patterns since the structure of attention maps is stable at different decoding steps [50, 78, 22, 42]. In Figure 15, we visualize the large attention map with hundreds of tokens on the PiQA [4] dataset to further verify our observation in Section 3. As can be observed, there are three types of tokens defined in Section 3: 1) IA tokens in red blocks which usually appear as attention sinks mentioned in StreamingLLM [76]. 2) IC tokens in orange blocks; 3) UIA tokens in blue blocks; The pattern of attention maps motivates us to statically discard the UIA tokens which may have negligible impact on further generation, and dynamically select important tokens from IC tokens at each decoding step for sparse attention computation.

## D  Pseudocode of MPCACHE Framework

We describe the overall flow of our MPCACHE in detail as shown in Algorithm 3.

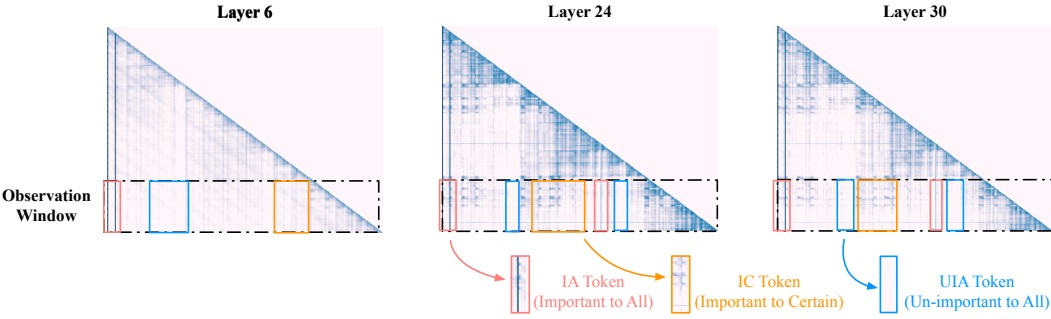

Figure 15: Attention patterns across different layers on LLaMA-7B.

---

**Algorithm 3:** MPCACHE: KV cache eviction combining static and dynamic algorithm

---

**Input** : Input sequence `prompt`; LLM model $\mathcal{M}$; number of layers and attention heads $L$ and $H$; dynamic selection ration $\eta \in [0, 1]$; three types of tokens IA, IC, and UIA (introduced in Section 3); cluster size $s$; decoding steps $E$.

**Output :** Attention output and evicted KV cache.

---

1 **Step 1: Look-once static eviction during prefill stage**
2 **for** $l \in [0, \ldots, L-1]$ **do**
3     $\mathbf{Q}^{(l)}, \mathbf{K}^{(l)}, \mathbf{V}^{(l)} \leftarrow \mathcal{M}(\text{prompt})$;
4     **if** $l\%2 == 0$ **then**
5        $\mathbf{ATTN}_{\text{window}}^{(l)} \leftarrow \text{Softmax}(\mathbf{Q}^{(l)}[:, -len(\text{prompt}) \times 0.2 :, :] \cdot \mathbf{K}^{(l)\top})$;
6        $\{IA, IC, UIA\}^{(l)} \leftarrow \text{static\_evict}(\mathbf{ATTN}_{\text{window}}^{(l)})$;
7     **else**
8        $\{IA, IC, UIA\}^{(l)} \leftarrow \{IA, IC, UIA\}^{(l-1)}$;        ▷ Cross-layer index sharing (Section 4.4)
9     $\mathbf{K}^{(l)} \leftarrow \text{token\_gather}(\mathbf{K}^{(l)}, index = \{IA, IC\}^{(l)})$;
10    $\mathbf{V}^{(l)} \leftarrow \text{token\_gather}(\mathbf{V}^{(l)}, index = \{IA, IC\}^{(l)})$;
11 **Step 2: Query-aware dynamic selection during decoding stage**
12 **for** $e \in [0, \ldots, E-1]$ **do**
13     **for** $l \in [0, \ldots, L-1]$ **do**
14        **if** $l\%2 == 0$ **then**
15           $\mathbf{sim} \leftarrow \text{Sim}(\mathbf{q}^{(l,e)}, \mathbf{K}^{(l,e)}, cluster\_size = s)$;        ▷ Follow Equation (3)
16           $\mathbf{index}^{(l,e)} \leftarrow \text{topk}(\mathbf{sim}, k = \mathbf{K}^{(l,e)}.size() \times \eta)$;
17        **else**
18           $\mathbf{index}^{(l,e)} \leftarrow \mathbf{index}^{(l-1,e)}$;        ▷ Cross-layer index sharing (Section 4.4)
19        $\mathbf{K}^{(l,e)} \leftarrow \text{token\_gather}(\mathbf{K}^{(l,e)}, index = \mathbf{index}^{(l,e)})$;
20        $\mathbf{V}^{(l,e)} \leftarrow \text{token\_gather}(\mathbf{V}^{(l,e)}, index = \mathbf{index}^{(l,e)})$;
21        $\mathbf{O}^{(l,e)} \leftarrow \text{Softmax}(\mathbf{q}^{(l,e)} \cdot \mathbf{K}^{(l,e)\top}/\sqrt{d}) \cdot \mathbf{V}^{(l,e)}$;        ▷ Sparse attention
22 **return** $\mathbf{O}, \mathbf{K}, \mathbf{V}$.

---

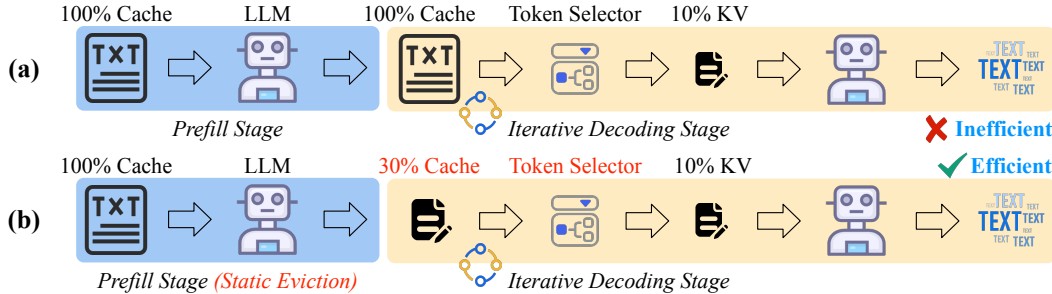

Figure 16: Paradigm comparison between (a) dynamic algorithm and (b) our proposed MP-CACHE combining static eviction and dynamic selection. MPCACHE discards unimportant tokens to reduce the decoding overhead (red texts mean the differences).

# E   Paradigm Comparison with Dynamic Algorithm

We compare the paradigm between the dynamic algorithm and our MPCACHE as shown in Figure 16. Through performing static eviction during the prefill stage, we improve the efficiency of all decoding steps since the token selector (i.e., similarity approximation and top-$k$ ranking) only needs to handle a smaller number of tokens during the decoding stage (30% in this case).

# F  Supplemental Experiments

## F.1  Supplemental Setups

**Experimental environment.** The model performance is evaluated with LongBench on an NVIDIA A100 80GB GPU in PyTorch. The latency is evaluated with Secretflow under the LAN setup [63] with 377MBps bandwidth and 0.3ms echo latency [63] on Intel(R) Xeon(R) Gold 5220R CPU @ 2.20GHz. To save GPU memory and avoid the out-of-memory (OOM) error when processing long contexts, we leverage FlashAttention [14] during the prefill stage. Since securely evaluating a full-size 7B model in SPU exceeds our hardware resources, we set a smaller embedding dimension of 1024 in our evaluation.

**Datasets.** In our experiments, we use XSUM [57], LongBench [2], and Needle-in-a-Haystack [23]. LongBench is a benchmark for bilingual, multitask, and comprehensive assessment of long context understanding capabilities of large language models. We also use the XSUM [57] dataset which is used for the evaluation of abstractive single-document summarization systems. Needle-in-a-Haystack is an evaluation method that randomly inserts key information into long texts to form prompts for LLMs. The test aims to detect whether models can extract such key information from extensive texts, thereby assessing the models' capabilities in processing and understanding long documents.

**KV cache clustering configuration.** For hierarchy, we in practice choose a two-level hierarchical structure, i.e., $n = 2$, and when the final dynamic selection ratio $\alpha < 0.5$, we drop 50% clusters at the 1st hierarchical level. For the XSUM dataset, we use a cluster size of 8 at the 1st hierarchical level and 4 at the 2nd hierarchical level. For the long-context LongBench, we use larger clusters, i.e., 32 at the 1st hierarchical level and 16 at the 2nd hierarchical level.

**Static eviction.** During the static eviction, we compute the accumulated attention sum only using the last 20% tokens in the prompt. However, using 20% tokens still incurs a CUDA out-of-memory (OOM) error when processing a long-context prompt (i.e., longer than 24k tokens). To solve this problem, we adaptively adjust the ratio to 10% tokens instead. We also notice that the choice of the ratio won't cause significant performance fluctuations, so we omit the discussion in this paper.

**How to securely determine the hyper-parameters, e.g., eviction ratios in the realistic scenario?** In practice, we believe the static and dynamic eviction ratio should be determined based on two aspects: 1) the expected sequence length and 2) the latency that can be tolerated/afforded by the server and the client. Our framework proposes a novel algorithm with configurable hyper-parameters to enable exploring the Pareto front of the LLM performance and efficiency. The framework has been validated across different benchmarks and different LLMs, which demonstrates its value and potential for practical use cases.

## F.2  Detailed Descriptions about Baselines

StreamingLLM [76] follows a fixed eviction pattern, i.e., keep the local tokens and initial tokens) across different decoding steps. H2O [91] and TOVA [58] statically prune the KV cache and these discarded tokens cannot be recovered at subsequent decoding steps. SnapKV [42] statically prunes the KV cache only during the prefill stage. InfLLM [74] employs block-level dynamic token selection during the decoding stage. It requires selecting several representative tokens within a block and computing the relevance score using these representative tokens. LongCache [47] uses the idea of cosine similarity between the query and key cache of previous tokens to select relevant tokens without static eviction and token clustering. Note that LongCache separates the positional embedding (PE) from the KV cache. Since our focus is on the dynamic selection metric in this work, we do not apply PE separation in LongCache.

## F.3  Supplemental Ablation Studies

**Effect of hyper-parameter $\alpha$.** To study how $\alpha$ impacts the similarity approximation, we select different $\alpha$'s on different datasets as shown in Figure 17. As can be observed, although the effects of different $\alpha$ do not occur in a certain pattern, we can still discover some patterns related to the dataset from the trend in Figure 17: on TriviaQA, the model may prefer larger $\alpha$ while it may prefer smaller $\alpha$ on HotpotQA instead. Since $\alpha = 0.6$ shows relatively better performance in these cases, we choose $\alpha = 0.6$ by default in our experiments.

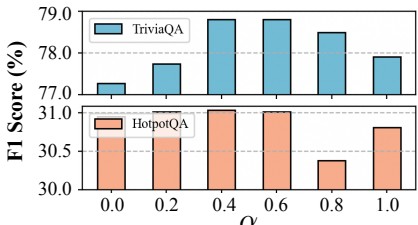

Figure 17: The influence trend of similarity approximation with different $\alpha$ values ranging from 0 to 1.

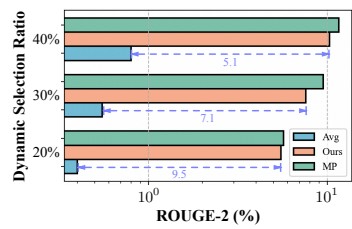

Figure 18: Comparison with average-based similarity approximation. MP means maximum dot product.

**Comparison with average-based similarity approximation.** A straightforward and efficient way to aggregate the information of a key cache cluster is the average. We compare our proposed method ($\alpha = 0.6$) with average-based similarity on the XSUM dataset with a cluster size of 16 in Figure 18. Specifically, we perform dynamic selection with different ratios after 75% tokens are statically discarded. As can be observed, using average suffers from significant performance degradation under different ratios. With the compression ratio increasing, the degradation of the average-based method becomes more serious. An intuitive explanation is that using the average of a cluster may make some important tokens averaged and ignored. In contrast, our approximation can effectively maintain the model performance. We theoretically analyze the similarity approximation algorithm in Appendix G.

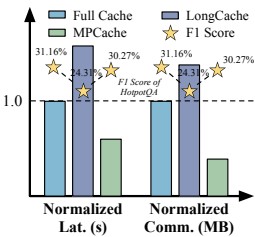

Figure 19: Extension MPCACHE to 2PC protocol.

**Discussion on 2PC protocol.** We evaluate the 2PC efficiency in Figure 19. It is observed that MPCACHE achieves $1.63\times$ and $1.79\times$ latency and communication reduction compared with the full cache, and $2.58\times$ and $2.48\times$ latency and communication reduction compared with LongCache. Since the multiplication communication in 2PC is larger than in 3PC, the cost of similarity approximation becomes higher. To solve this problem, we can leverage the random projection based on Johnson-Lindenstrauss (JL) Lemma [34] to reduce the dimensionality while preserving the token distance. We leave the research as our future work.

**Static eviction ratio.** We show the ablation study to explain our choice of static eviction ratio in Table 7. We choose the static eviction ratio that does not significantly affect the performance on the datasets, and different static eviction ratios can influence the overall decoding performance.

## G   Theoretical Analysis of Similarity Approximation

As mentioned in Section 2, given query $\mathbf{q} \in \mathbb{R}^{H \times 1 \times d}$, key cache $\mathbf{K} \in \mathbb{R}^{H \times T \times d}$, and value cache $\mathbf{V} \in \mathbb{R}^{H \times T \times d}$, the overall goal of KV cache eviction is to find an optimal policy $\mathcal{P}$ to minimize the gap between the attention outputs (here we omit $\mathbf{V}$ for simplification) as

$$\mathcal{P}^* = \text{argmin}|\text{Softmax}(\mathbf{q} \cdot \mathbf{K}^\top) - \text{Softmax}(\mathbf{q} \cdot \mathbf{K'}^\top)|, \quad (7)$$

where $\mathbf{K'}$ is a subset of $\mathbf{K}$ selected by $\mathcal{P}$. However, when grouping $\mathbf{K}$ into clusters for efficiency, the problem becomes challenging. We denote the key cache cluster as $\mathbf{K}_c$ (cluster size is $s$), and our goal is to find a way to accurately approximate the similarity between $\mathbf{q}$ and the key cluster $\mathbf{K}_c$. This problem is equivalent to *"how can we effectively aggregate the cluster information to obtain a cluster representation and measure its importance?"*

Table 7: Model performance with different static eviction ratios on HotpotQA.

| Eviction Ratio (%) | F1 Score (%) |
|---|---|
| 0 | 31.16 |
| 55 | 30.82 |
| 70 | 30.54 |
| 75 | 28.89 |

We assume there exists a function $\phi$ that aggregates the key cluster $\mathbf{K}_c$, and we define the optimization problem as

$$\min |\sum_{j=0}^{s-1} \exp(\mathbf{q} \cdot \mathbf{K}_{cj}^\top) - \mathbf{q} \cdot \phi(\mathbf{K}_c^\top)|. \quad (8)$$

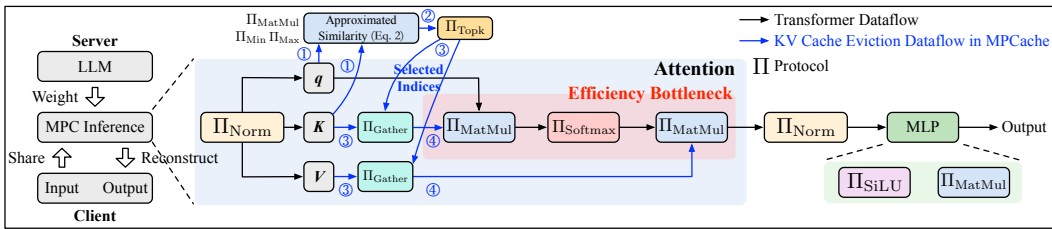

Figure 20: Overall secure inference data flow of MPCACHE during token generation.

**Drawback of average-based clustering.** As mentioned, the simplest way to represent the cluster is the average and $\phi(\mathbf{K}_{cj}^\top)$ becomes $\sum_{j=0}^{s-1} \mathbf{K}_{cj}^\top / s$. This happens to be equivalent to directly drop $\exp$ in $\sum_{j=0}^{s-1} \exp(\mathbf{q} \cdot \mathbf{K}_{cj}^\top)$, introducing information loss. *Intuitively, if there are tokens with low importance and tokens with high importance within one cluster, the overall importance will be averaged, leading to the neglect of the crucial tokens.*

Different from the average-based method, using the max dot product aims to protect the crucial tokens with large scores as much as possible. This observation is aligned with [67].

$$\text{MaxDotProduct}: \quad \mathbf{q} \cdot \phi(\mathbf{K}_c^\top) = \max_{\mathbf{k} \in \mathbf{K}_c} \mathbf{q} \cdot \mathbf{k}. \tag{9}$$

In order to approximate $\max_{\mathbf{k} \in \mathbf{K}_c} \mathbf{q} \cdot \mathbf{k}$ without accessing all the tokens in $\mathbf{K}_c$, we utilize the bounding volume proposed by [36]. Our optimizations are described in Section 4.3.

## H  Overall Secure Inference Framework

Take 2PC as an example in Figure 20, we illustrate the secure inference framework following [53] where the server owns the proprietary LLM parameter and the client possesses private input data. During inference, the data is secretly shared between two parties. Linear layers are computed using the HE protocol, and non-linear layers require interactive protocols between the two parties based on oblivious transfer (OT) and HE. Figure 20 illustrates the detailed data flow of MPCACHE during the token generation.

