# OpenReview forum: "MPCache: MPC-Friendly KV Cache Eviction for Efficient Private LLM Inference"
_NeurIPS.cc/2025/Conference — NeurIPS 2025 poster_

### Official Review · Reviewer_5USt · 2025-06-09

**Clarity:** 3
**Significance:** 2
**Originality:** 2
**Rating:** 5
**Confidence:** 2

**Summary:**

The paper proposes MCPCache, a novel KV cache eviction strategy for use in LLM inference when performed via MPC. They thoroughly review and compare the state of the art and justify why their strategy works.

**Questions:**

Please address how the speed tradeoffs here fit into the limitations of the broader approach.

**Ethical Concerns:**

["NO or VERY MINOR ethics concerns only"]

**Final Justification:**

The authors have vigorously and thoroughly responded to all the reviewer comments.

While there are still challenges with the underlying pragmatic utility of this scope of research, they have argued for its novelty and soundness.

I am happy to increase my review to an acceptable (from borderline acceptable) and retain my confidence.

**Limitations:**

No, see above.

**Quality:**

3

**Strengths And Weaknesses:**

It's a good paper, with a clear framework for why it is designed the way it is, and an explanation of how speed is improved via the KV eviction strategy.

Stregths:
- Clear explanation of why the strategy works
- The extensive experiments demonstrate significant improvements over existing methods. MPCACHE achieves a reduction of up to 2.01x latency and 8.37x communication.

Weaknesses:
- No Explicit Limitations Section: The paper does not have a dedicated "Limitations" section, and the authors state in the checklist that the paper has no limitations. MPC fundamentally has limitations due to communication complexity and isn't widely used in AI deployments for that reason.
-More generally, it's unclear with these speedups if anyone would be willing to accept the complexity of MPC inference.
- Assumption of Architectural Knowledge: The security model assumes that all parties are aware of the model architecture and the number of pruned tokens. This is fine and aligns with past literature, but it forgoes significant privacy.

---

> ### Author Rebuttal · Authors · 2025-07-26
>
> We sincerely thank Reviewer 5USt for your thoughtful feedback!
>
> ---
>
> **Q1:** No explicit limitations section and address how the speed tradeoffs fit into the limitations of the broader approach.
>
> **A1:** Thanks for your valuable advice! We agree that MPC incurs more communication overhead and is still under research, but we believe MPC has strong potential for these reasons: 1) In the era of LLM, the issue of data privacy becomes more severe, and there is a strong demand for LLM's privacy protection. For example, General Data Protection Regulation (GDPR) provides policy-level protections for LLM's data privacy; 2) MPC provides cryptographically-strong privacy protection for both data and parameters and is also independent of the hardware platforms; 3) In recent years, there is a significant amount of research on MPC inference, especially for LLMs [1-10].
>
> Through our optimizations, **MPCache achieves up to 2.01$\times$ latency and 8.37$\times$ communication reduction.** In addition, MPCache is **orthogonal to both protocol and hardware,** and thus can be integrated with future advanced protocols and hardware to further enhance usability. For example, according to Piranha [11], GPU can accelerate MPC by 16-48$\times$.
> Therefore, **we believe this research has important academic value and can promote the practical application of MPC.**
>
> In our revised version, we will add the limitation section to elaborate the gap between private inference and the plaintext counterpart.
>
> ---
>
> **Q2:** Assumption of architectural knowledge.
>
> **A2:** Thanks for your thoughtful consideration. Our threat model is consistent with previous studies that the model architecture and pruning ratio are public. We believe this is reasonable in real-world applications, where users typically know the model architecture and pruning ratio (or speedup) when using the service. For example, **Gemini [12] discloses the model architecture and optimizations while keeping model parameters private.**
>
> Importantly, our adopted threat model does not compromise the client's data, nor does it enable the client to access the model's parameters, which is **aligned with the security goal of MPC.**
>
> ---
>
> **References:**
>
> [1] Zhang, Yancheng, et al. "CipherPrune: Efficient and Scalable Private Transformer Inference." ICLR 2025.
>
> [2] Lu, Wen-jie, et al. "Bumblebee: Secure two-party inference framework for large transformers." NDSS 2025.
>
> [3] Li, Zhengyi, et al. "Nimbus: Secure and Efficient Two-Party Inference for Transformers." NeurIPS 2024.
>
> [4] Pang, Qi, et al. "Bolt: Privacy-preserving, accurate and efficient inference for transformers." 2024 IEEE Symposium on Security and Privacy (SP). IEEE, 2024.
>
> [5] Rathee, Deevashwer, et al. "MPC-Minimized Secure LLM Inference." arXiv preprint arXiv:2408.03561 (2024).
>
> [6] Zheng, Fei, et al. "PermLLM: Private Inference of Large Language Models within 3 Seconds under WAN." arXiv preprint arXiv:2405.18744 (2024).
>
> [7] Li, Dacheng, et al. "Mpcformer: fast, performant and private transformer inference with mpc." ICLR. 2023.
>
> [8] Zeng, Wenxuan, et al. "Mpcvit: Searching for accurate and efficient mpc-friendly vision transformer with heterogeneous attention." Proceedings of the IEEE/CVF International Conference on Computer Vision. 2023.
>
> [9] Hou, Xiaoyang, et al. "Ciphergpt: Secure two-party gpt inference." Cryptology ePrint Archive (2023).
>
> [10] Dong, Ye, et al. "Puma: Secure inference of llama-7b in five minutes." arXiv preprint arXiv:2307.12533 (2023).
>
> [11] Watson, Jean-Luc, Sameer Wagh, and Raluca Ada Popa. "Piranha: A GPU platform for secure computation." 31st USENIX Security Symposium (USENIX Security 22). 2022.
>
> [12] Comanici, Gheorghe, et al. "Gemini 2.5: Pushing the frontier with advanced reasoning, multimodality, long context, and next generation agentic capabilities." arXiv preprint arXiv:2507.06261 (2025).

---

> ### Author Response · Authors · 2025-08-09
> **Follow-up on Discussion for Paper 10762**
>
> Dear Reviewer 5USt,
>
> I hope this message finds you well.
>
> We sincerely appreciate your recognition of our work. As the discussion period is nearing its end with less than 6 hours, we want to ensure we have addressed all your concerns satisfactorily. If there are any additional feedback you'd like us to consider, please let us know.
>
> Thank you once again for your time and effort in reviewing our paper.
>
> Sincerely,
>
> The Authors of Paper 10762

---

### Official Review · Reviewer_kcBh · 2025-06-23

**Clarity:** 3
**Significance:** 2
**Originality:** 3
**Rating:** 4
**Confidence:** 3

**Summary:**

The authors mainly address the high latency issue in securely performing KV cache eviction. To reduce the cost of similarity computation, they propose cache clustering, where similarity is calculated at the cluster level and refined progressively in a coarse-to-fine manner. To improve effectiveness, instead of using the mean of each cluster, they measure similarity using the maximum and minimum key values within the cluster. Additionally, they further enhance efficiency by exploiting the similarity of eviction patterns across adjacent layers, allowing similarity computation to be performed only on selected layers. Experiments conducted on various datasets and architectures demonstrate that the proposed method is both more efficient and effective compared to existing methods.

**Questions:**

* The reduction in communication cost is significant, but the corresponding latency acceleration appears to be relatively limited. Could the authors provide an analysis explaining this discrepancy?
* Figures 7 and 8 appear after Figure 9 and are placed far from the text where Figure 7 is first mentioned. It would be better to relocate them closer to their first reference for improved readability.

**Ethical Concerns:**

["NO or VERY MINOR ethics concerns only"]

**Final Justification:**

While the latency improvement is somewhat limited, the method is novel and meaningful in that it reduces both latency and memory usage. Therefore, I have raised my score to 4: borderline accept.

**Limitations:**

The paper does not explicitly address its limitations.

**Paper Formatting Concerns:**

No formatting issues were found.

**Quality:**

3

**Strengths And Weaknesses:**

### Strengths
* The authors propose a novel and efficient method using cache clustering.
* They conduct extensive experiments to demonstrate the effectiveness of the proposed method.

---

###  Weaknesses
* The proposed method appears to achieve approximately a 2× speedup in Figure 10. However, according to Figure 3(a), static eviction also shows no performance degradation up to 60%. Therefore, when compared to static eviction methods applied at a 60% rate, the proposed method might offer similar performance but little to no advantage in terms of speed improvement, since static eviction approaches are more efficient than dynamic eviction. Therefore, a comparison with static eviction at a 50–60% ratio in terms of both latency and performance is necessary.
* Since the method requires additionally storing the minimum and maximum values for each cluster, it is expected to incur higher memory usage. In large models with long sequence lengths, this overhead could become significant. It would be helpful to include a comparison of memory consumption to assess this impact.
* While the improvement in latency over existing methods is understandable, the reason for the performance improvement is not convincing at a high level. It would be helpful if the authors could provide a high-level analysis explaining why the proposed method outperforms the dynamic KV cache baseline in terms of effectiveness.

---

> ### Author Rebuttal · Authors · 2025-07-29
>
> We sincerely thank Reviewer kcBh for your thoughtful feedback!
>
> ---
>
> **Q1:** Comparison with static eviction at a 50–60% ratio in terms of latency and performance.
>
> **A1:** Thank you for the insightful comment. First, we would like to clarify that the latency of Transformer consists of attention and MLP, and KV cache eviction only optimizes attention. **The 2$\times$ latency reduction shown in Figure 10 represents an end-to-end result, and thus, it is not equivalent to 50% static eviction.**
>
> In fact, achieving the **~2$\times$ latency reduction requires compressing the KV cache to less than 10%**. As shown in the table below, **solely** **applying 60% static eviction leads to limited efficiency improvement (i.e., only 1.21$\times$** **latency and 1.83$\times$ communication** **reduction**), which is significantly lower than MPCache. Meanwhile, using only static eviction to reduce latency by ~2$\times$ would result in substantial accuracy degradation (this can be also observed in SnapKV in Figure 9).
>
> | Method (Reserved KV Cache) | Latency (s) | Communication (GB) | Accuracy (%) |
> | --- | --- | --- | --- |
> | Full KV Cache (100%) | 75.52 | 1.82 | 26.54 |
> | Pure Static Eviction (40%) | 62.12 (1.21$\times$) | 0.99 (1.83$\times$) | 25.56 |
> | Pure Static Eviction (8%) | 40.32 (1.87$\times$) | 0.36 (5.06$\times$) | 16.80 **(Significant Drop)** |
> | MPCache (8%) | 44.39 (1.70$\times$) | 0.41 (4.44$\times$) | 23.86 |
>
> In contrast, MPCache builds upon static eviction and further perform dynamic selection to identify the most important KV cache. This allows for **more aggressive compression, especially for communication cost (i.e., 1.7$\times$ latency and 4.4$\times$ communication reduction) while maintaining accuracy**, achieving significantly higher accuracy than using static eviction alone.
>
> We will include these clarifications and the comparison results in our revised version.
>
>
> ---
>
> **Q2:** This work requires additionally storing the min and max for each cluster, it is expected to incur higher memory usage.
>
> **A2:** Thanks for your thoughtful consideration. We would like to clarify that the memory usage introduced by storing min/max values is **negligible compared to the KV cache.** Specifically, KV cache has a dimension of $(H, T, D)$, where $H$, $T$, and $D$ are head number, token number, and hidden dimension per head, respectively. In contrast, min/max values are stored per cluster, resulting in a much smaller dimension of $(H, T/s, D)$, where $s$ denotes cluster size. It means min/max values require only $1/s$ memory compared to KV cache. In our experiments, the cluster size $s$ is very large, i.e., 32, resulting in only $1/32$ of the KV cache size.
>
> For example, for a Llama-2-7B model and a sequence length of 16K, storing KV cache requires 7.8 GB, while storing min and max values requires only 250 MB of storage (in FP16 precision), which is ~3% of KV cache size.
>
> Since the **min/max value requires minimal storage, and communication and running time are the dominant bottleneck in current MPC protocols**, we prioritize communication and latency reduction in this work. We will add more explanations regarding the memory usage in our revised version.
>
> ---
>
> **Q3:** The reason for the performance improvement is not convincing at a high level.
>
> **A3:** We analyze the effectiveness of MPCache at a high level as follows:
>
> - **Comparison with average-based clustering methods.** The widely adopted dynamic KV cache method is to represent each cluster using average of all tokens within it. However, this method suffers from the drawback of **averaging out the token importance,** making it difficult to retain highly informative tokens given a KV cache budget. In contrast, **MPCache is guided by the principle that similarity approximation should preserve the influence of important tokens within each cluster**. So, we build a novel similarity approximation method that utilizes the maximum dot product between the query and key clusters, rather than the average, to better approximate token importance. This intuition is empirically validated across different models and datasets, which shows that our design improves the selection of important tokens under a constrained KV cache budget.
>
> - **Comparison with non-clustering (token-wise) methods.** In long-context scenarios, prior studies has observed attention noise [1-3], where the dot product between the query and key becomes less reliable in reflecting token importance when the sequence is very long. Hence, compared to these token-wise similarity approximation methods, MPCache utilizes cluster-wise granularity to reduce the approximation complexity and **mitigate the attention noise** to some extent, enabling more robust and accurate token selection, especially for long sequences.
>
> In conclusion, MPCache improves performance by using a novel similarity approximation that effectively preserves clusters with influential tokens, which is especially useful for long-context scenarios. We will clarify these motivations and high-level insights in the revised version.
>
> ---
>
> **Q4:** The reduction in communication cost is significant, but the corresponding latency acceleration is relatively limited.
>
> **A4:** The primary reason for this discrepancy is that the **communication scales proportionally with the input dimension,** resulting in a significant decrease when the KV cache is compressed. In contrast, **latency is influenced by multiple factors,** including computation time, the number of communication rounds in MPC, and network setups such as bandwidth and transmission delay. These components **do not scale linearly with input size**, thereby limiting the overall latency reduction.
>
> Furthermore, as our current implementation runs on CPU, leveraging GPU to accelerate computation in future work could yield higher latency gains.
>
> We will clarify these points and include this discussion in our revised version.
>
> ---
>
> **Q5:** It would be better to relocate the figures closer to the first reference for improved readability.
>
> **A5:** Thanks for your kind advice! We will adjust the locations of figures and texts carefully in our revised version.
>
> ---
>
> **Lastly, thanks so much for helping us improve this work through your valuable feedback. If you have any additional questions or anything you would like to discuss further, please feel free to let us know. If you find that we have addressed your concerns, please kindly consider re-evaluating our work. Thank you once again for your time and effort!**
>
> ---
>
> **References:**
>
> [1] Ye, Tianzhu, et al. "Differential transformer." arXiv preprint arXiv:2410.05258 (2024).
>
> [2] Adnan, Muhammad, et al. "Keyformer: Kv cache reduction through key tokens selection for efficient generative inference." Proceedings of Machine Learning and Systems 6 (2024): 114-127.
>
> [3] Wang, Xue, et al. "S3attention: Improving long sequence attention with smoothed skeleton sketching." IEEE Journal of Selected Topics in Signal Processing (2024).

---

### Official Review · Reviewer_suRu · 2025-06-26

**Clarity:** 2
**Significance:** 3
**Originality:** 3
**Rating:** 4
**Confidence:** 4

**Summary:**

This paper presents an MPC-friendly KV cache eviction framework. MPC is a cryptography scheme that allows multiple parties to work together on the computation with leaking the plaintext. However, some operations are not efficient in MPC in terms of computation latency and communication, such as comparison and non-linear operations. The existing KV cache eviction on plaintext computation is not efficient in MPC. This paper studies the KV cache eviction in the context of MPC. The proposed scheme combines static eviction and dynamic selection. It reduces the expensive max and similarity approximation operations needed, and also has optimizations like hierarchical clustering and cross-layer index-sharing strategy. The evaluation shows the proposed eviction outperforms the baseline KV cache eviction mechanism in both model performance and latency.

**Questions:**

* Please explain the color coding in Figure 1.

* What is the implementation like in Figure 1(b) and (c)? Is KV cache used? If so, what KV cache eviction algorithm is used?

* Is the latency in Table 3 per inference or for X number of queries? How the numbers are measured are not clearly stated in the paper not the evaluation.

* Can you provide a further breakdown in "model" Figure 11? Earlier in the paper, you mentioned that softmax accounts for the majority of the cost. How about after your KV cache optimizations?

* Can you provide insights on what the performance will be like if the networking has longer latency or bandwidth?

**Ethical Concerns:**

["NO or VERY MINOR ethics concerns only"]

**Final Justification:**

Thanks for the response. My clarification questions are addressed. For practical use, the concerns remain, and I will keep my score. In general, I still support the paper.

**Limitations:**

I agree with the authors that the proposed scheme is general. However, the resulting MPC is still slow for practical use. A discussion on MPC for decoder models can be added.

**Quality:**

3

**Strengths And Weaknesses:**

Strengths
+ (Significance and Originality) The first study on KV cache eviction in the MPC framework. This work will make the decoder models more feasible in MPC. I like the proposed schemes that reduce the number of max operations.
+ (Quality) The proposed optimizations are derived from observations in detailed performance profiling. The paper is well-written and supported by solid experiments.

Weaknesses
- (Quality) The paper can provide more details on the performance improvement due to each of the optimizations. The current results do not give insights into the impact of each optimization.
- (Quality) The evaluation metrics and how it is measured are not stated.
- (Quality) The evaluation does not test under different network setups.
If the authors cannot evaluate the framework under different network settings, the computation and communication breakdown (i.e., percentage of time in communication vs. computation, or are they pipelined?) should be provided at least.

- (Clarity) Some figures miss necessary legends.
-- For example, Figure 1 did not mark what each color means in the breakdown.
-- In Figure 1(d), the KV cache choice of the eviction baseline [47] can be biased. Table 3 shows that baseline [47] has worse latency performance than baseline InfLLM.

-- It should also be stated clearly that F1 score is used as the model performance metric.

---

> ### Author Rebuttal · Authors · 2025-07-27
>
> We sincerely thank Reviewer suRu for your thoughtful feedback!
>
> ---
>
> **Q1:** Provide more details on the performance improvement due to each of the optimizations.
>
> **A1:** Thanks for you advice! We here summarize  the impact of each optimization from the perspective of accuracy and efficiency.
>
> - Static eviction algorithm: unimportant KV cache is pruned based on the attention scores with negligible overhead. It does not affect the prefill stage and remain partial KV cache after the prefill stage to **improve the overall efficiency by ~60% without degrading accuracy.**
> - Dynamic selection algorithm: 1) similarity approximation optimization: a clustering-based optimization is proposed and it preserves the impact of important tokens within a cluster as much as possible. Hence, our selection method is **accurate and efficient (further improve overall efficiency by ~60%);** 2) linearization & reordering and hierarchical clustering optimizations: they aim to improve the efficiency of similarity approximation by removing MPC-expensive operators and progressive selection **(reduce KV cache selection overhead by ~80%) without significantly affecting accuracy;** 3) cross-layer index sharing optimization: due to the high similarity between adjacent layers as observed in Section 3, this optimization directly **reduces the overhead of KV cache selection by 50% without degrading accuracy.**
>
> In conclusion, our optimizations are supported either by theoretical analysis or by empirical observations. These optimizations work together to improve the efficiency while preserving accuracy. **Experimental analysis can be found in our step-by-step ablation studies in Section 5.4.** We will add more discussions in our revised version.
>
> ---
>
> **Q2:** The evaluation metrics and how it is measured are not stated.
>
> **A2:** Thanks for your suggestion! F1 score used in LLM generation task is calculated as the harmonic mean of precision and recall at the token level: F1=2$\times$Precision$\times$Recall/(Precision+Recall), where precision measures the proportion of tokens in the generated output that also appear in the answer, and recall measures the proportion of tokens in the answer that are correctly included in the generated output.
>
> In this paper, we follow LongBench [11] to measure the metrics, and we will state the metric and measurement details clearer in our revised version.
>
> ---
>
> **Q3:** The evaluation does not test under different network setups.
>
> **A3:** Thanks for your valuable advice! We provide the breakdown of computation vs. communication latency to better understand the protocols. We evaluate the 3PC protocol on Llama-2-7B to generate 1 token with a sequence lengths of 512, as shown in the table below:
>
> | Network | Total Lat. (s) | Comm. (GB) | Comm. Lat. (s) | Comput. Lat. (s) | **+ MPCache** |
> | --- | --- | --- | --- | --- | --- |
> | LAN | 51.2 | 1.01 | 2.679 (5.23%) | 48.521 (94.77%) | **25.5s (2.0$\times$) / 0.24GB** |
> | WAN | 73.8 | 1.01 | 25.25 (34.2%) | 48.521 (65.8%) | **31.2s (2.3$\times$) / 0.24GB** |
>
> As observed, under different network setups, communication improvement remains unchanged. In contrast, latency improvement can vary depending the network setups. For WAN with lower bandwidth, the improvement can be slightly higher, i.e., 2.3x since KV cache eviction directly reduces the communication volume.
>
> We will include more comprehensive experiments and analysis about the network setups in our revised version.
>
> ---
>
> **Q4:** Some figures miss necessary legends and explain the color coding in Figure 1.
>
> **A4:** Thanks for pointing out our oversights! We have carefully revised the writing to make our paper clearer. In Figure 1, blue bar means attention, orange bar means MLP, green bar means LayerNorm, purple bar in slash means similarity approximation, yellow bar in slash means top-k ranking, and green bar in slash means token gathering.
>
> ---
>
> **Q5:** Implementation like in Figure 1(b) and (c). Is KV cache used and what KV cache eviction algorithm is used?
>
> **A5:** Figure 1(b) uses the both 2PC and 3PC protocols on GPT-2 using the Secretflow framework. Figure 1(c) uses the 3PC protocol to evaluate Softmax. KV cache is used here to avoid redundant re-computation of KV cache (refer to Appendix A.3 for details). But in Figure 1(b) and (c), KV cache eviction algorithm is not used since they are profiled to analyze the efficiency bottleneck with full cache, which motivate us to compress the KV cache.
>
> We will carefully improve our writing to make the paper clearer in our revised version.
>
> ---
>
> **Q6:** Latency evaluation in Table 3.
>
> **A6:** The latency in Table 3 is measured by generating one token on Llama-2 given one input with a sequence length of 1024 tokens. We will clarify all the measurements in our revised version.
>
> ---
>
> **Q7:** Provide a further breakdown in "model" Figure 11.
>
> **A7:** We show the model-level communication breakdown with a sequence length of 1024 in the table below. As observed, before optimization, Softmax dominates the total cost. After optimization, Softmax overhead becomes very small, leading to significant overall efficiency improvement. We will include these experiments in our revised version.
>
> | Method | Softmax (MB) | MLP (MB) | Others (MB) | Total (MB) |
> | --- | --- | --- | --- | --- |
> | Full Cache | 1242.7 | 184.32 | 436.66 | 1863.68 |
> | + MPCache | 68.19 | 184.32 | 23.97 | 276.48 |
>
> ---
>
> **Q8:** The resulting MPC is still slow for practical use. A discussion on MPC for decoder models can be added.
>
> **A8:** Thanks for your valuable advice! We agree that MPC incurs more communication overhead and is still under research, but we believe MPC has strong potential. MPC has been used in practical applications such as healthcare [12] and finance [13]. In recent years, there is a significant amount of research on MPC inference for LLMs, which is more complex and expensive [1-10]. Through our optimizations, **MPCache achieves up to 2.01$\times$ latency and 8.37$\times$ communication reduction.**
>
> In addition, MPCache is orthogonal to both protocol and hardware, and thus can be integrated with future advanced protocols and hardware to further enhance usability. For example, according to Piranha [14], GPU can accelerate MPC by 16-48$\times$. Therefore, **we believe this research has important academic value and can promote the practical application of MPC.**
>
> In our revised version, we will add more discussions and the limitation section regarding the MPC-based decoder models.
>
> ---
>
> **References:**
>
> [1] Zhang, Yancheng, et al. "CipherPrune: Efficient and Scalable Private Transformer Inference." ICLR 2025.
>
> [2] Lu, Wen-jie, et al. "Bumblebee: Secure two-party inference framework for large transformers." NDSS 2025.
>
> [3] Li, Zhengyi, et al. "Nimbus: Secure and Efficient Two-Party Inference for Transformers." NeurIPS 2024.
>
> [4] Pang, Qi, et al. "Bolt: Privacy-preserving, accurate and efficient inference for transformers." 2024 IEEE Symposium on Security and Privacy (SP). IEEE, 2024.
>
> [5] Rathee, Deevashwer, et al. "MPC-Minimized Secure LLM Inference." arXiv preprint arXiv:2408.03561 (2024).
>
> [6] Zheng, Fei, et al. "PermLLM: Private Inference of Large Language Models within 3 Seconds under WAN." arXiv preprint arXiv:2405.18744 (2024).
>
> [7] Li, Dacheng, et al. "Mpcformer: fast, performant and private transformer inference with mpc." ICLR. 2023.
>
> [8] Zeng, Wenxuan, et al. "Mpcvit: Searching for accurate and efficient mpc-friendly vision transformer with heterogeneous attention." Proceedings of the IEEE/CVF International Conference on Computer Vision. 2023.
>
> [9] Hou, Xiaoyang, et al. "Ciphergpt: Secure two-party gpt inference." Cryptology ePrint Archive (2023).
>
> [10] Dong, Ye, et al. "Puma: Secure inference of llama-7b in five minutes." arXiv preprint arXiv:2307.12533 (2023).
>
> [11] Bai, Yushi, et al. "Longbench: A bilingual, multitask benchmark for long context understanding." arXiv preprint arXiv:2308.14508 (2023).
>
> [12] Ahammed, Md Fahim, and Md Rasheduzzaman Labu. "Privacy-preserving data sharing in healthcare: advances in secure multiparty computation." Journal of Medical and Health Studies 5.2 (2024): 37-47.
>
> [13] Salako, Ademola Oluwaseun, et al. "Securing confidentiality in distributed ledger systems with secure multi-party computation for financial data protection." Journal of Engineering Research and Reports 27.3 (2025): 352-373.
>
> [14] Watson, Jean-Luc, Sameer Wagh, and Raluca Ada Popa. "Piranha: A GPU platform for secure computation." 31st USENIX Security Symposium (USENIX Security 22). 2022.

---

> > ### Comment · Reviewer_suRu · 2025-08-06
> >
> > Thanks for the response. My clarification questions are addressed. For practical use, the concerns remain, and I will keep my score. In general, I still support the paper.

---

### Official Review · Reviewer_mW9c · 2025-06-29

**Clarity:** 3
**Significance:** 3
**Originality:** 3
**Rating:** 5
**Confidence:** 3

**Summary:**

This work proposes a framework for properly integrating KV caching into MPC-based LLM inference to reduce inference costs. To do this, they use an MPC-friendly similarity approximation, hierarchical KV cache clustering, and a cross-layer index-sharing strategy to reduce the overhead of dynamic token selection. Their experimental results demonstrate strong performance and efficiency results compared to existing KV cache schemes.

**Questions:**

- Is obersvation 1 a general observation, not specific to the MPC setting?
- Is this work the first to address optimization in MPC-based LLM inference with KV cache?

**Ethical Concerns:**

["NO or VERY MINOR ethics concerns only"]

**Final Justification:**

After reading through the other reviewers' and rebuttals, I will still keep my score. The paper presentation is easy to follow, and the core ideas/contributions are interesting. Their rebuttal addressed many of the concerns raised by other reviewers and should be incorporated into their final version. Hence, I am generally supportive of the work.

**Limitations:**

yes

**Paper Formatting Concerns:**

None.

**Quality:**

3

**Strengths And Weaknesses:**

Strengths:
- Tackles an important problem of revising the KV cache scheme to be more MPC-friendly
- The writing style of the paper is great, with clear descriptions about the problem setup, motivational observations, and the proposed method
- Lots of figures are utilized in the main text that help provide insight into important claims
- The ideas proposed in the MPC-friendly dynamic KV cache selection section of the paper seem very interesting and innovative
- Code is open-source and available for reproducibility
- Experimental evaluations are comprehensive, on both performance and efficiency, with the proposed method comparable or outperforming existing static and dynamic KV cache methods

Weaknesses:
I have no major complaints about the work. I only have some minor complaints that may need to be addressed:
- A legend is needed for Figure 1, as there are colors used in some of the graphs, and I do not know what they correspond to.
- The descriptions for the figures and tables should be more comprehensive and self-contained. For example, for Figure 9 and Table 3, I do not know what models are being evaluated on and what performance metrics are being used.
- For the tables, it would be visually easier to bold face the numbers that are best amongst the others
- It would be insightful to include a non-KV cache baseline for both the performance and efficiency baselines

---

> ### Author Rebuttal · Authors · 2025-07-26
>
> We sincerely thank Reviewer mW9c for your appreciation of our work and thoughtful feedback!
>
> ---
>
> **Q1:** Legend missing in Figure 1.
>
> **A1:** Thanks for pointing this out! In Figure 1, blue bar means attention, orange bar means MLP, green bar means LayerNorm, purple bar in slash means similarity approximation, yellow bar in slash means top-k ranking, and green bar in slash means token gathering. We have added the legend in our revised version.
>
> ---
>
> **Q2:** The descriptions for the figures and tables should be more comprehensive.
>
> **A2:** Thanks for your kind advice! Figure 9 and Table 3 are evaluated on LongChat-7B-V1.5-32K on the LongBench [1] benchmark. We will carefully improve the descriptions for the figures and tables in our revised version.
>
> ---
>
> **Q3:** For the tables, it would be visually easier to bold face the numbers that are best.
>
> **A3:** Thanks for your suggestion! We will improve the tables to make the results clearer.
>
> ---
>
> **Q4:** It would be insightful to include a non-KV cache baseline for both the performance and efficiency baselines.
>
> **A4:** Thank your for your valuable advice! For performance, non-KV cache is consistent with the full cache, since the purpose of the KV cache is to reduce repeated computation (please refer to Appendix A.3 for details).
>
> For efficiency, compared with using KV cache, non-KV cache baseline incurs hundreds of times latency, communication, and memory. Here, we provide an example: non-KV cache baseline requires more than 800GB of communication with a input sequence length of 2048. This comparison demonstrates that KV cache drastically reduces the overhead and further highlights the efficiency gains achieved by MPCache.
>
> | Method | Communication (GB) | Latency (s) |
> | --- | --- | --- |
> | Non-KV Cache Baseline | 829.4 | 18144 |
> | + KV Cache (Full Cache) | 3.460 | 111.4 |
> | + MPCache | 0.413 | 61.86 |
>
> We promise to add comprehensive results of the non-KV cache baseline in our revised version.
>
> ---
>
> **Q5:** Is observation 1 a general observation, not specific to the MPC setting?
>
> **A5:** Yes. Observation 1 focuses on the KV cache sparsity analysis and classification, which motivates us to combine static and dynamic KV cache compression for LLM efficiency improvement.
>
> ---
>
> **Q6:** Is this work the first to address optimization in MPC-based LLM inference with KV cache?
>
> **A6:** Yes. This work is the first research to introduce KV cache for MPC-based LLM inference. Also, this is the first to study long-context inference in MPC. In this paper, to alleviate the heavy overhead introduced by long contexts (i.e., large KV cache size), we further propose a series of MPC-friendly optimizations for KV cache compression to improve the overall efficiency while preserving the accuracy.
>
> ---
>
> **References:**
>
> [1] Bai, Yushi, et al. "Longbench: A bilingual, multitask benchmark for long context understanding." arXiv preprint arXiv:2308.14508 (2023).

---

### Official Review · Reviewer_AHwV · 2025-07-03

**Clarity:** 3
**Significance:** 2
**Originality:** 2
**Rating:** 4
**Confidence:** 5

**Summary:**

The paper presents MPCache, a kKV cache eviction framework optimized for multi-party computation (MPC)-based private LLM inference. It combines static eviction during the prefill stage to prune unimportant tokens. Also, MPCache is incorporated with dynamic, query-aware cluster-based selection to further reduce the KV cache at each decoding step.

**Questions:**

See weaknesses.

**Ethical Concerns:**

["NO or VERY MINOR ethics concerns only"]

**Final Justification:**

My concern has been addressed. I have raised my score. Please include the updated experiments in the camera-ready version to ensure the effectiveness of the approach.

**Limitations:**

See weaknesses.

**Paper Formatting Concerns:**

No.

**Quality:**

3

**Strengths And Weaknesses:**

Strengths:
1. The paper introduces a hybrid static + dynamic KV eviction strategy tailored to MPC constraints
2. This is a training-free model that can be directly applied on existing LLMs

Weaknesses:
1. Optimizations are largely heuristic and empirical, lacking formal approximation bounds or convergence analysis.
2. Generally speaking, the experiment setup is not sufficient. For example, efficiency section $5.3$ Fig.10, only MHA models like Llama-2 are used, which gives a limited demonstration of how efficient MPCache is on SOTA models.
3. In the section $4.4$, the authors missed a citation when discussing index sharing across adjacent layers. This finding has been extensively discussed in the paper: [https://arxiv.org/abs/2410.05076]{https://arxiv.org/abs/2410.05076}.

---

> ### Author Rebuttal · Authors · 2025-07-28
>
> We sincerely thank Reviewer AHwV for your thoughtful feedback!
>
> ---
>
> **Q1:** Optimizations are largely heuristic and empirical, lacking formal approximation bounds or convergence analysis.
>
> **A1:** Thank you for your insightful comment. For the **similarity approximation,** we have provided rigorous derivation in Eq. (1)-(2) and theoretical analysis in Appendix G, which demonstrate that MPCache better preserves the clusters with large $qK$ values, i.e., important tokens, and thus, achieves higher accuracy. This analysis is grounded in the concept of bounding volumes from computer graphics [1-4], which we adapt to formally analyze and approximate token importance under constrained KV cache budgets.
>
> For other optimizations proposed by MPCache:
>
> - **Cross-layer index sharing** is motivated by inter-layer consistency patterns, as observed in Figures 3 and 8.
> - **Hierarchical clustering** utilizes the tree structure to effectively balance efficiency and accuracy. We provide the computational complexity analysis: let $T$ be the total token number, $c_1$ and $c_2$ be the cluster number at the 1st and 2nd level,  and $k$ be the preserved token number. Therefore, the complexity of full cache is $O(T)$, and hierarchical clustering only requires $O(c_1+c_2+k)$, where $c_1, c_2, k \ll T$ in our experiments.
>
> In conclusion, while some of our optimizations are not derived from formal theory, they are grounded in empirical observations which are **consistently verified across a wide range of datasets and models, leading to strong practical credibility.** We will incorporate more formal analysis in the revised version to further strengthen our work.
>
> ---
>
> **Q2:** The experiment setup is not sufficient. In Figure 10, only MHA models like Llama-2 are used, which gives a limited demonstration of how efficient MPCache is on SOTA models.
>
> **A2:** Thank you for the suggestion. **MPCache is generally applicable to different model architectures like Grouped Query Attention (GQA)** used in recent SOTA models, e.g., Llama-3 and Qwen-3. GQA reduces KV storage and the number of linear projections (e.g., $XW^K$), but **the computation of attention—$\text{Softmax}(qK^\top)V$—remains the same as MHA**, since each query still attends to all keys and values within its group.
>
> Since our method is designed for MPC, **only computation reduction leads to lower communication and latency, whereas storage savings do not.** As a result, the benefits of MPCache naturally extend to both MHA and GQA. In the table below, we add results on Llama-3-8B with GQA at different sequence lengths. **The result shows significant latency and communication improvements, which is consistent with MHA.**
>
> We will supplement our experimental setup and incorporate these results into the revised version.
>
> | **Sequence Length** | **512 (Lat. /Comm.)** | **1024 (Lat. /Comm.)** | **2048 (Lat. /Comm.)** |
> | --- | --- | --- | --- |
> | Full Cache | 50.1s / 0.98GB | 73.1s / 1.76GB | 108.2s / 3.46GB |
> | LongCache (5% Cache) | 56.5s / 0.97GB | 86.3s / 2.03GB | 154.9s / 4.94GB |
> | ArkVale (5% Cache) | 37.3s / 0.42GB | 55.6s / 0.54GB | 90.02s / 0.73GB |
> | MPCache (5% Cache) | 26.4s / 0.24GB | 38.9s / 0.30GB | 62.54s / 0.42GB |
> | **Improvement over Full Cache** | **1.89$\times$ / 4.08$\times$** | **1.87$\times$ / 5.86$\times$** | **1.73$\times$ / 8.23$\times$** |
>
> ---
>
> **Q3:** The authors missed a citation.
>
> **A3:** Thank you for your kind suggestion! We will include the related paper in our revised version.
>
> ---
>
> **Lastly, thanks so much for helping us improve this work through your valuable feedback. If you have any additional questions or anything you would like to discuss further, please feel free to let us know. If you find that we have addressed your concerns, please kindly consider re-evaluating our work. Thank you once again for your time and effort!**
>
> ---
>
> **References:**
>
> [1] Efficient Collision Detection of Complex Deformable Models using AABB Trees. Journal of Graphics Tools, 1997.
>
> [2] Collision Detection Using Axis Aligned Bounding Boxes. Simulations, Serious Games and Their Applications. Springer Singapore, Singapore, 2014.
>
> [3] Bounding volume. Wikipedia, April 2024.
>
> [4] Efficient collision detection using bounding volume hierarchies of k-DOPs. IEEE transactions on Visualization and Computer Graphics 4.1 (1998)

---

> > ### Comment · Reviewer_AHwV · 2025-08-05
> >
> > Thanks for authors' replies to my review. My concern has been addressed. I have raised my score. Please include the updated experiments in the camera-ready version to ensure the effectiveness of the approach.

---

> > > ### Author Response · Authors · 2025-08-05
> > > **Thanks very much for your support - Response to Reviewer AHwV**
> > >
> > > Dear Reviewer AHwV,
> > >
> > > Thanks so much for your support and willingness to raise the score! We are glad to know that we have addressed your concerns.
> > >
> > > We truly appreciate your thoughtful engagement throughout the review process, and we will incorporate all the discussions and experiments into our final version.
> > >
> > > Sincerely,
> > >
> > > The Authors of Paper 10762

---

### Note · Authors · 2025-08-12

Dear NeurIPS 2025 ACs, SACs, PCs, and Reviewers,

We would like to sincerely thank all five reviewers for their time, effort, and the **unanimous recognition of our work,** as well as the ACs, SACs, and PCs for their dedicated service.

**Summary of positive feedback:**
We deeply appreciate the reviewers' positive feedback, highlighting our paper's clear writing (Reviewer mW9c, 5USt, suRu), the novelty and creativity of our proposed method (Reviewer AHwV, mW9c, suRu, kcBh), the broad applicability (Reviewer AHwV), and the thoroughness of our experimental evaluation (Reviewer mW9c, suRu, kcBh, 5USt).

**Summary of comments and rebuttal:**
The reviewers primarily provided suggestions regarding the writing and figures, as well as recommendations to include additional experimental setups. For example, evaluations on alternative attention mechanisms (e.g., GQA), comparisons with non-KV cache baseline, and experiments under different network configurations. During the rebuttal period, we have addressed these comments by providing all the suggested experiments and reporting the corresponding results. **We are glad that our responses addressed each reviewer's concerns, and Reviewer AHwV and kcBh decided to raise their scores.**

**Significance of this work:**
In the era of LLM, the issue of data privacy becomes more severe, and there is a strong demand for LLM's privacy protection. To reduce the huge overhead of private LLM inference, we propose an accurate and efficient KV cache eviction framework with a combination of static and dynamic algorithms. To enable efficient dynamic KV cache selection, our framework further incorporates a series of optimizations, including MPC-friendly similarity approximation, hierarchical clustering, and cross-layer index-sharing strategy. Extensive experiments demonstrate the significant speedup over various baselines.


Once again, we express our heartfelt gratitude for the constructive feedback and recognition from the reviewers, and for the ACs, SACs, and PCs' hard work in overseeing the review process.

Sincerely,

The Authors of Paper 10762

---

### Decision · Program_Chairs · 2025-09-17

**Decision:**

Accept (poster)

**Comment:**

The paper introduces a framework called MPCache. It is designed to make private (in MPC sense) large language model (LLM) inference faster. While secure multi-party computation (MPC) protect user data, it is known to be slow. MPCache speeds up MPC by a novel approach to the key-value (KV) caching. It combines a static method to discard unimportant cache data and a dynamic method to select only a small, relevant subset of the cache for each step.

Reviewers found the hybrid approach of MPCach to be a novel and important contribution, as it is the first to address KV cache eviction within the MPC framework. The method is also "training-free", meaning it can be applied directly to existing LLMs.  Initially, the reviewers raised several concerns. These included an insufficient experimental setup, such as a lack of newer models and tests under different network conditions. They also asked for more details on memory usage, clearer figures, and a better explanation of the performance gains. The authors' response addressed most of these points, leading several reviewers to raise their scores. There are still concerns about the practiacality of the approach, but overall the reviewers find the paper a step forward toward making MPC LLMs efficient. The authors should also include the new experiments and discussions in their revision.